# Highly multiplexed selection of RNA aptamers against a small molecule library

**Brent Townshend**[1☯¤], **Matias Kaplan**[1☯¤], **Christina D. Smolke**[1,2]*

**1** Department of Bioengineering, Stanford University, Stanford, CA, United States of America, **2** Chan Zuckerberg Biohub, San Francisco, CA, United States of America

☯ These authors contributed equally to this work.

¤ Current address: Atomic AI, South San Francisco, CA, United States of America

* csmolke@stanford.edu

**Data Availability Statement:** All relevant data are within the manuscript and its Supporting Information files.

**Funding:** This work was supported by the National Institutes of Health (NIH grant R01 GM 086663 to

## Abstract

Applications of synthetic biology spanning human health, industrial bioproduction, and ecosystem monitoring often require small molecule sensing capabilities, typically in the form of genetically encoded small molecule biosensors. Critical to the deployment of greater numbers of these systems are methods that support the rapid development of such biosensors against a broad range of small molecule targets. Here, we use a previously developed method for selection of RNA biosensors against unmodified small molecules (DRIVER) to perform a selection against a densely multiplexed mixture of small molecules, representative of those employed in high-throughput drug screening. Using a mixture of 5,120 target compounds randomly sampled from a large diversity drug screening library, we performed a 95-round selection and then analyzed the enriched RNA biosensor library using next generation sequencing (NGS). From our analysis, we identified RNA biosensors with at least 2-fold change in signal in the presence of at least 217 distinct target compounds with sensitivities down to 25 nM. Although many of these biosensors respond to multiple targets, clustering analysis indicated at least 150 different small-molecule sensing patterns. We also built a classifier that was able to predict whether the biosensors would respond to a new compound with an average precision of 0.82. Since the target compound library was designed to be representative of larger diversity compound libraries, we expect that the described approach can be used with similar compound libraries to identify aptamers against other small molecules with a similar success rate. The new RNA biosensors (or their component aptamers) described in this work can be further optimized and used in applications such as biosensing, gene control, or enzyme evolution. In addition, the data presented here provide an expanded compendium of new RNA aptamers compared to the 82 small molecule RNA aptamers published in the literature, allowing further bioinformatic analyses of the general classes of small molecules for which RNA aptamers can be found.

## Introduction

Molecular components that support sensing are critical to many biological systems. Fitness is often contingent on responding to the presence and concentration of chemicals in an

C.D.S.), National Science Foundation (graduate fellowships to M.K.), and Howard Hughes Medical Institute (Gilliam graduate fellowship to M.K.). C.D. S. is a Chan Zuckerberg Biohub investigator. The funders had no role in study design, data collection and analysis, decision to publish, or preparation of the manuscript.

**Competing interests:** Brent Townshend and Christina Smolke are co-inventors of patent US20200362333A1: "Molecular Sensor Selection". Matias Kaplan declares no competing interests exist.

organism's environment. Natural biological systems have evolved a diversity of sensor types and corresponding mechanisms. Furthermore, small molecule sensing capabilities are critical to applications of synthetic biology which span human health, industrial bioproduction, and ecosystem monitoring [1, 2]. As the field explores greater numbers of these engineered biological systems, methods that can support the scalable and rapid development of new biosensors that can detect diverse small molecules are critical.

The field has developed a number of different molecular platforms for developing small molecule biosensors, including engineered transcription factors, enzymes, and nucleic acid aptamers; however, methods described to-date generally require an extensive application-specific development cycle for new biosensor components [3–7]. An ideal system for developing small molecule biosensors would incorporate a well-understood platform that can be used to rapidly screen, either *in silico* or *in vitro*, for sensors capable of sensing a diverse range of small molecule targets and be easily tethered to an actuator component that supports both *in vivo* or *in vitro* readout [1, 2].

Early work in RNA biochemistry led to the development of methods such as Systematic Evolution of Ligands by EXponential enrichment (SELEX) for the *in vitro* selection of ligand-binding RNA sequences, or aptamers, from large libraries of random RNA sequences [8–10]. Since the original description of SELEX, improvements to support more rapid selection approaches and to enable the generation of aptamers with greater specificities and affinities have been described. These include changes in library design, selection strategies, incorporation of modified or unnatural nucleotides, and computational modeling of selection techniques [11–14]. Despite these advances the number of ligands that can be sensed by nucleic acid aptamers remains relatively low, with 168 total small molecule ligands that can be sensed by nucleic acid aptamers reported as of 2017, 82 of which use RNA as the sensor [13, 15, 16].

Recent work from our laboratory demonstrated a method called *de novo* rapid *in vitro* evolution of RNA biosensors (DRIVER), which was successfully used to create new small molecule biosensors to six diverse small molecules that previously did not have a sensor [17]. DRIVER utilizes aptamer-coupled ribozyme libraries and relies on sequence changes in the ribozyme following cleavage to select for ligand-sensitive cleavage (Fig 1A). Specifically, DRIVER relies on a unique ribozyme regeneration step following cleavage to support efficient and unbiased regeneration of active ribozymes in the pool to enable solution-based separation of RNA biosensors. Further detail for DRIVER is provided in the results section. We also developed and validated CleaveSeq, a high-throughput parallelized assay based on NGS, to characterize new biosensors in parallel by counting cleaved and uncleaved reads for each biosensor sequence in mixed biosensor libraries. The biosensors selected through DRIVER exhibit nanomolar to micromolar sensitivities and were also shown to directly function *in vivo* in yeast and mammalian cell systems to regulate gene expression with up to 33-fold activation ratios [18]. Gene expression can be controlled by placing the ligand-responsive ribozymes in the 3'-UTR of a target mRNA; when the ribozymes cleave, they separate the eukaryotic poly-A tail from the rest of the transcript, thereby targeting the transcript for degradation and lowering gene expression.

In this work we explored the utility of DRIVER to be a rapid and efficient generator of new small molecule biosensors to diverse small molecule compounds. We performed a DRIVER selection against a library of 5,120 diverse small molecule target compounds that were selected from a high throughput drug screening library. The compound library was assembled into mixtures for selection, and the library itself was verified pre- and post-mixing using liquid-chromatography quantitative time of flight mass spectrometry (LC/QTOF-MS). After 95 rounds of selection on the DRIVER platform, 334 RNA sequences were identified as possible biosensors. The small molecule targets of those potential biosensors were subsequently

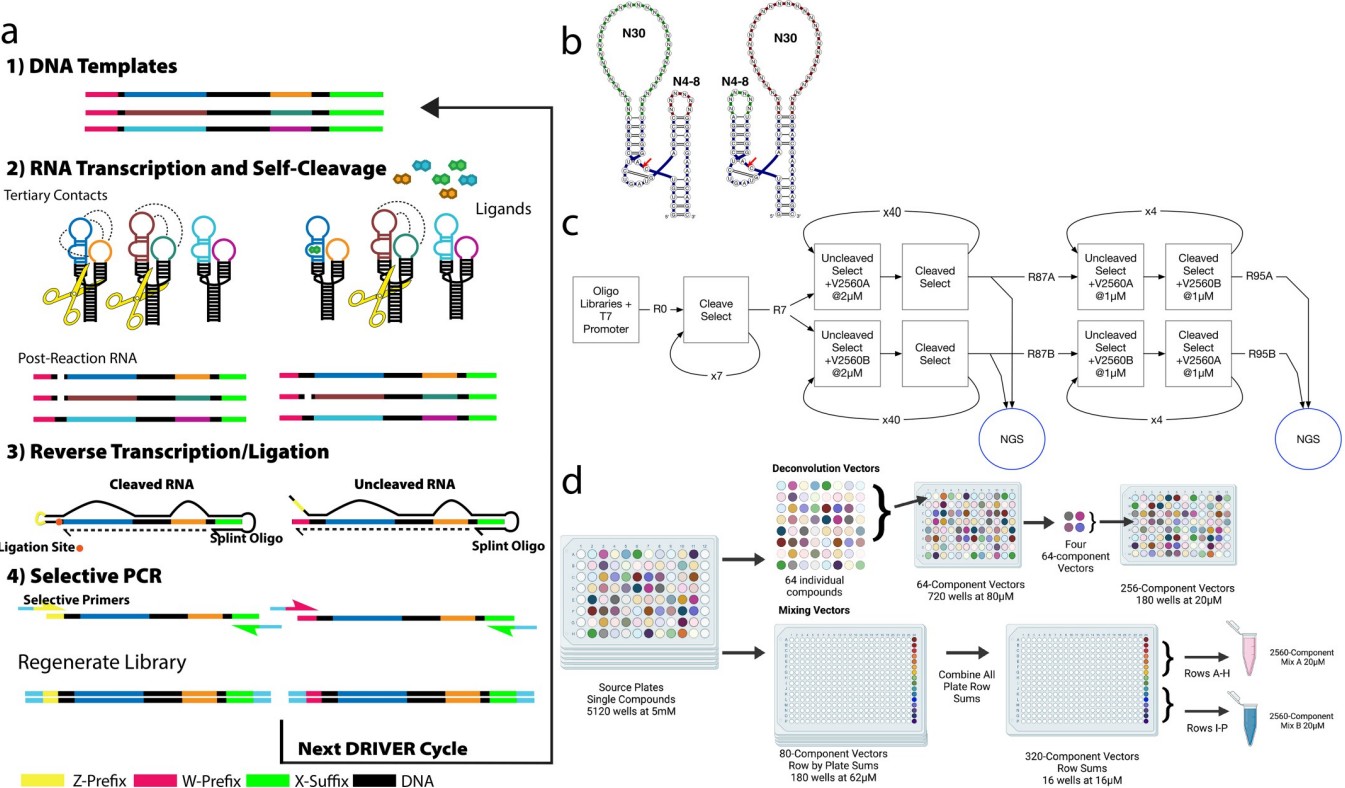

**Fig 1. DRIVER overview and small molecule library setup. (a)** Overview of DRIVER process. Sequences are transcribed in the absence or presence of ligands and allowed to self-cleave. An individual sequence is then either in a cleaved or uncleaved state. At this stage the RNAs are mixed with a splint oligonucleotide whose 3' end acts as a reverse transcription (RT) primer. Following RT, the splint oligonucleotide's 5' end can anneal to the 3' end of the cDNA corresponding to the cleaved sequences such that efficient ligation of a new prefix occurs for the uncleaved sequences. Following RT and ligation, the two prefixes can be used to distinguish between cleaved and uncleaved sequences–either for library regeneration or for quantification. **(b)** Secondary structure representation of general RNA biosensor library design with the loop randomizations indicated. N6 small loops and N30 large loops are shown. **(c)** DRIVER selection was performed for 95 rounds of selection followed by NGS analysis of products using CleaveSeq. **(d)** Source plates containing 5μl per well of 5120 compounds at 5μM in DMSO were reformatted to form two selection mixtures, V2560A & V2560B, and 180 256-component mixtures (V256-1-1 to V256-9-20).

identified by assessing the activities of potential biosensor sequences against a set of orthogonal vector mixtures of the small molecule target library with CleaveSeq [19]. We then validated the ligand responsiveness of these biosensors in the presence of individual small molecule targets, resulting in identification of 217 small molecule targets that produce at least 2-fold change in cleavage activity in response to ligand in one or more of the identified RNA biosensors.

## Results and discussion

### Hammerhead ribozyme-based biosensor selection library

The RNA biosensor library was designed to create a high-diversity library ($10^{12}$–$10^{14}$) that could produce small-molecule-modulated, self-cleaving RNA sequences to new target ligands. The biosensor design is based on the satellite RNA of tobacco ringspot virus (sTRSV) hammerhead ribozyme (Fig 1B) [20]. The sTRSV hammerhead ribozyme consists of three helices and two loops surrounding a core. It is postulated that under physiological low $Mg^{2+}$ concentrations, tertiary interactions between the loops stabilize the core, which allows the ribozyme to adopt a catalytically active form thereby leading to self-cleavage. In our library design, one loop is replaced with a randomized 30 nucleotide region intended to give rise to aptamer sequences, while the other loop was replaced with 4 to 8 nucleotide random region intended to

produce tertiary interactions with the sequence on the opposite loop. We have previously shown that the presence of the aptamer's cognate ligand can interfere with these interactions and result in modulation of self-cleavage of the ribozyme [17, 21]. The ribozyme-based biosensor sequence is flanked by a 5' T7 RNA polymerase promoter and an A-rich sequence ("W Prefix") and a 3' distinct A-rich sequence ("X Suffix"). These two flanking sequences are used for manipulating the library sequences, serving as PCR handles and, by the presence or absence of the W prefix, distinguishing between cleaved and uncleaved sequences. The RNA biosensor library is synthesized as DNA oligonucleotides in the antisense direction and annealed to a T7 promoter oligonucleotide to support T7 RNA runoff transcription of the template for the synthesis of the corresponding RNA biosensor library.

## Automated DRIVER selection allows for multi-round enrichment of RNA sensors against target compound libraries

DRIVER selection was performed on the RNA biosensor library over multiple rounds to progressively enrich for sequences that exhibit low self-cleavage in the presence of the target compound library, defined as positive selection, and high self-cleavage in the absence of the compound library, defined as negative selection. DRIVER cycles consist of four main steps: 1) transcription of the sequences, 2) self-cleavage of the transcripts, 3) reverse-transcription of the transcripts, 4) ligation of a new prefix to cleaved 3' fragments, and 5) a selective PCR (Fig 1A). For either selection round RNA is transcribed through an enzymatic *in-vitro* transcription. During positive selection this transcription occurs in the presence of ligands, while in negative selection no target ligands are present. Following incubation, any individual RNA sequence will either be intact or have undergone self-cleavage which removes the prefix sequence. A novel splint oligonucleotide is combined with the RNA sequences and used as a reverse transcription primer. Following reverse transcription, cDNA corresponding to cleaved sequences are ligated to the splint oligonucleotide, replacing the prefix that was removed by cleavage with a different prefix. The process allows the introduction of sequences with differing prefixes, which can be used for selective PCR amplification. During positive selection cycles, PCR is performed with the PCR primers to keep uncleaved sequences while negative selection cycles incorporate PCR primers that only amplify cleaved sequences. The cycles then alternate between positive and negatives selections in order to enrich for biosensors–sequences that cleave in the absence of ligand and do not cleave in the presence of ligand.

The DRIVER selection was performed by beginning with seven rounds of selection for cleaving sequences in the absence of the target compound library. This initial enrichment was performed to bias the starting RNA biosensor library toward high-cleaving ribozyme sequences, as biosensor sequences that do not cleave in the absence of ligand are unlikely to exhibit a high fold-activation. After this initial enrichment, alternating rounds of positive selection (i.e., selection of non-cleaving sequences in the presence of the target compound library) and negative selection (i.e., selection of cleaving sequences in the absence of the target compound library) were performed. Selection after Round 7 was performed in parallel on two independent series: one using V2560A as the target compound library mixture during non-cleaving rounds, the other using V2560B as the target compound library mixture. The selection was then performed for 80 alternating rounds with the target library mixture at 2 μM per compound. Finally, 8 additional alternating rounds of selection were performed where the non-cleaving rounds used the same target mixtures at a concentration of 1 μM per compound, and the cleaving rounds used the alternate V2560 mixture at 1 μM per compound. The end rounds of selection were designed to improve the selectivity of generated biosensors by de-

enriching sequences selected to respond to V2560A components that were also sensitive to components of V2560B and vice versa.

As all steps in DRIVER require only liquid movements and thermocycling, the DRIVER selection process is automated on a liquid-handling robot that can run continuously multiple rounds/day. However, selection was performed manually for the first four rounds due to the large solution volumes needed to maintain diversity prior to enrichment. Subsequent rounds were performed on an automated liquid-handling system which performed nine rounds of selection per day. After the initial manual rounds, the enriched biosensor libraries from each round were retained and in intervals of ~16 rounds the concentration of the enriched libraries were checked via qPCR to verify that the concentration stayed approximately constant, but the selection was otherwise run blind.

## Prototype small molecule target compound library designed to mimic drug and biologically relevant molecules

Testing the limits of DRIVER required us to build a target compound library comprising diverse small molecule targets that are representative of the breadth of small molecules for which biosensors might be desired. The target compound library comprises 5,120 small molecule compounds randomly selected from a ChemDiv representative diversity library obtained via the Stanford High-Throughput Bioscience Center [22]. The target compounds ranged in molecular weight from 112 to 500 Daltons (S1) and were supplied in 5 mM DMSO. The target compound library was reformatted from the initial set of 16 plates to 2 non-overlapping mixtures of 2,560 compounds each ("V2560A" and "V2560B") and 9 sets of 20 non-overlapping mixtures of 256 compounds each ("V256-1.01" to "V256-9.20") (Fig 1D). Each of the target compounds in the 256-compound mixtures was chosen randomly with the constraint that no mixture contained multiple target compounds with overlapping expected m/z mass spectra. The 2,560-compound mixtures were concentrated by evaporation of DMSO to 20 μM. The concentrated mixtures were then further diluted 3x with water and precipitated compounds were pelleted and removed from the mixtures to reduce any undesired target compound precipitation that might occur during the DRIVER selection steps.

The target compound libraries were validated by mass spectrometry to ensure that the expected compounds were present following the processing steps to build these libraries. One hundred of the V256 mixtures, which included each compound in five different mixtures, were analyzed on an Agilent 6545 Quantitative Time of Flight (QTOF) mass spectrometer. For each compound, the five V256 mixtures which were expected to contain that compound were analyzed along with five additional randomly selected mixtures that should not contain that particular compound. The data were compared to identify, as possible, a particular adduct and retention time that uniquely correspond to the compound of interest with minimal false positives or false negatives. The analysis indicated that one plate of 80 compounds was incorrect, and subsequent analysis indicated that the plate in question had been mislabeled at some point prior to this work and contained the contents of the adjacently numbered plate from the original high-throughput screening collection. The list of compounds used was updated to resolve this issue without loss of any data. Using this method, over 90% of the compounds (4,477 of 5,120) were identified (S2 Fig and S1 Table), providing validation that the expected compounds were indeed present in the mixtures used for selection and analysis. We postulate that the remaining compounds that were not identified through this method, comprising ~12% of the target compound libraries used in this study, either did not ionize in positive mode electrospray ionization or did not produce ion counts above the noise floor of the instrument.

## Multi-stage CleaveSeq analysis of DRIVER-enriched libraries reveals new biosensor sequences

The enriched RNA biosensor library generated by DRIVER was subsequently characterized using CleaveSeq [17, 19] to measure the relative cleavage activity of each individual sequence in the library in the absence of the target compound mixtures and in the presence of each of the target compound mixtures (V2560A, V2560B). For each condition, the RNA biosensor library was transcribed to RNA, where each sequence underwent self-cleavage at the conditions of the assay depending on the particular RNA sequence and target compounds present. The RNA sequences were then reverse-transcribed and cDNA corresponding to cleaved sequences was ligated to a prefix sequence distinct from that carried by the uncleaved sequences. The resulting sequences were barcoded, prepared as Illumina libraries, and were then sequenced. Counts of the reads corresponding to cleaved and uncleaved products arising from each library sequence were used to compute the cleavage fraction and fold change of cleavage for each sequence under each assay condition using the following formulas:

$$\text{Cleavage Fraction} = \frac{\# \text{ of reads cleaved}}{\# \text{ of reads uncleaved} + \# \text{ of reads cleaved}}$$

$$\text{Fold Change of Cleavage} = \frac{\# \text{ of reads cleaved}_{\text{without target}} \times \# \text{ of reads uncleaved}_{\text{with target}}}{\# \text{ of reads uncleaved}_{\text{without target}} \times \# \text{ of reads cleaved}_{\text{with target}}}$$

The CleaveSeq analysis indicated that 334 RNA sequences exhibited a fold change of cleavage of at least two in the presence of one or both of the target compound libraries (in each case passing a test of statistical significance with $p < 1/N$; (Fig 2)).

A synthesized RNA biosensor pool was designed based on results from the CleaveSeq analysis of the DRIVER-enriched biosensor libraries. Specifically, the 334 sequences identified as potential biosensors based on the CleaveSeq analysis and additional sequences that were present at high abundance in either of the enriched RNA biosensor libraries, were resynthesized using an oligonucleotide array. In all, this synthesized RNA biosensor pool contained 1,730 sequences. Of these, 168 sequences with high fold changes of cleavage were selected as "high-interest" sequences. The high-interest sequences were mixed in the pool with a 10x higher abundance than the other sequences. Details of the pool selection criteria are contained in S4 Table. Briefly, sequences were chosen that were either: suspected hits from sequencing selection rounds, high-abundance sequences, or suspected amplicon sequences. CleaveSeq characterization was performed on the synthesized RNA biosensor pool under various conditions and analysis was performed by initial shallow sequencing on an Illumina iSeq instrument. This approach provided enough reads for characterization of the sequences in the high-interest pool against the set of small molecule vectors. Selected conditions were then re-analyzed at a higher sequencing depth on an Illumina NextSeq to allow characterization of the complete synthesized RNA biosensor pool at these conditions, while improving statistics for the high-interest sequences as described in the next section.

## Pooled target compound testing and deconvolution identifies 217 new small molecule biosensors

The CleaveSeq characterization of the RNA biosensor libraries indicate which sequences have biosensor activity to compounds within the V2560A or V2560B target compound mixtures, but do not indicate to which compounds in those mixtures. Performing characterization assays against each of the 5,120 possible target compounds would be infeasible. Therefore, we took a

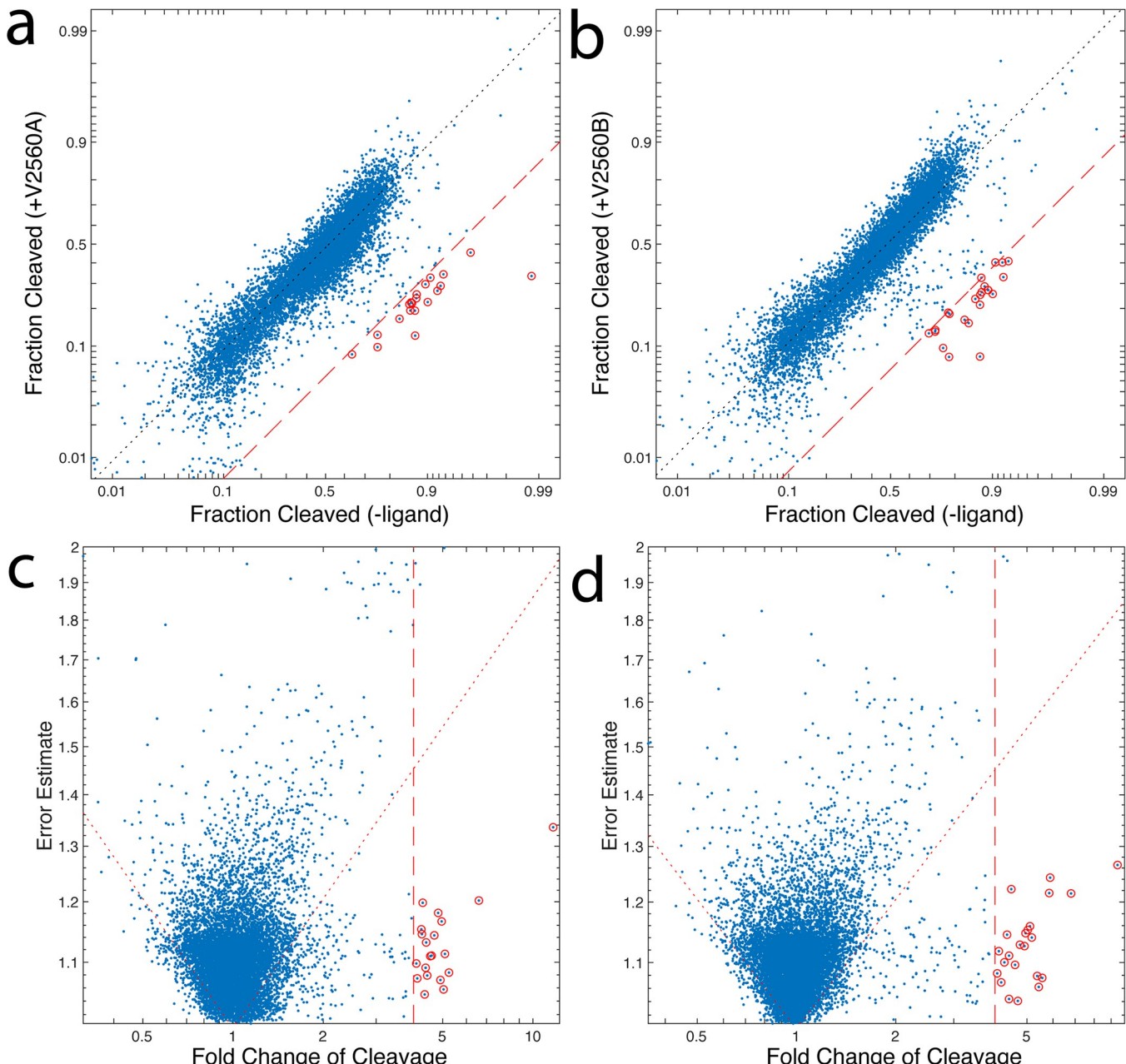

**Fig 2. Identification of statistically significant and high fold-chance of cleavage biosensor hits from DRIVER 5120.** Comparison of cleavage fractions for products of round 95 with and without target mixtures as determined using CleaveSeq. Left panels (**a,c**) show the response of each sequence to mixture V2560A and the right panels (**b,d**) to mixture V2560B. (N ~ 10000 sequences, at least 100 reads/sequence in each analysis). The top panels (**a,b**) show the cleavage of each measured sensor in each condition and the bottom panels (**c,d**) show the standard error of the cleaved:uncleaved read count ratio vs. the fold change of cleavage. Significant (two-sided test with Bonferroni correction: $p < 1/N$) outliers are shown with red circles. Red dashed lines delineate 4-fold change of cleavage. Dotted red lines in bottom panels show the threshold of significance ($p = 1/N$).

two-phase approach to identify the compounds that interact with each sequence of the synthesized RNA biosensor pool.

In the first phase, the synthesized pool was characterized using the CleaveSeq assay in the presence of each of the 180 256-compound mixtures, V256-{1–9}-{1–20} (Fig 3). The resulting data were analyzed to identify likely target compounds that would give rise to the observed

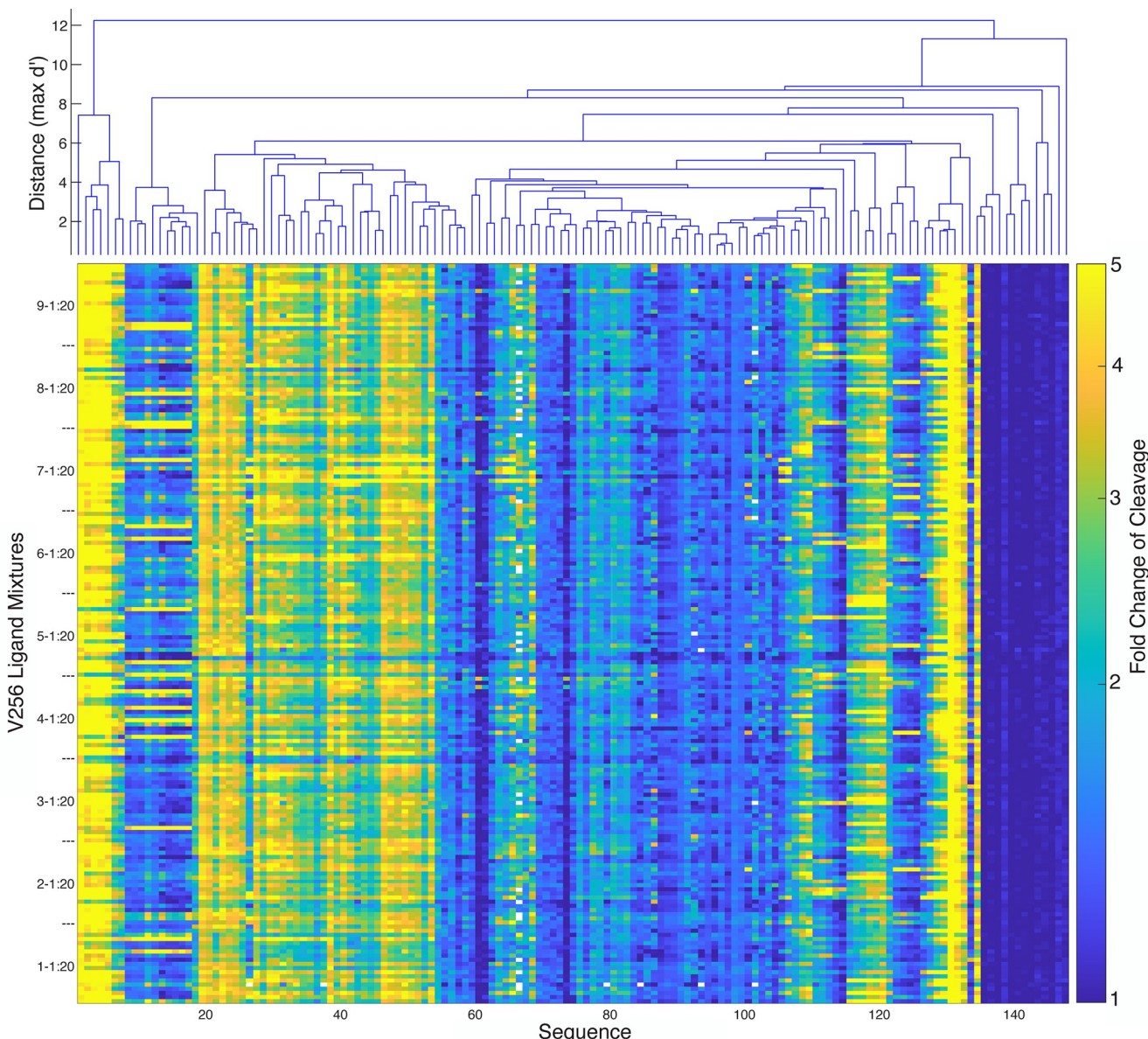

**Fig 3. CleaveSeq results of select biosensors against deconvolution vectors shows clusters of similar biosensors.** The pseudocolor plot shows the fold change of cleavage for each of 147 sequences (the most-frequent representative of clusters of sequences that showed similar response patterns) in the presence of each of the 180 mixtures of 256 compounds, each at 2μM concentration. Sequences are ordered based on hierarchical clustering of the patterns of response with the dendrogram above showing the response similarities. Dendrogram distances (d') are the maximum (over all 180 vectors) of the log of the ratio of fold changes divided by the standard deviation of the estimates.

patterns of fold change of cleavage. For example, biosensor 566229815 had a fold change of cleavage of more than 2 only in the presence of the V256 mixtures that contained compound 167A08, so it was highly likely that this compound was the cognate ligand for this biosensor. For most of the biosensor sequences, several of the V256 mixtures resulted in a response, and analysis identified which components were shared between the mixtures but not present in the mixtures that did not show an observable response. Biosensor sequences responsive to less than approximately 10 distinct target compounds could be characterized in this way. Although the vectors were designed to be orthogonal, if a sequence was responsive to more than 10

distinct target compounds, then a positive signal would be seen in most if not all the V256 mixtures. This led to insufficient information to deconvolute which specific molecules or even how many different molecules the promiscuous sequences were sensing. Withholding those sequences, we successfully deconvolved sensors for at least 217 different target compounds that at least one RNA biosensor exhibits greater than two-fold change of cleavage against (S5 Table). Due to our inability to deconvolve all possible biosensor-ligand pairs from the vector data this is a lower bound on the number of small molecules from the library that the DRIVER-generated biosensors are able to sense.

In the second phase, CleaveSeq assays were performed on the synthesized RNA biosensor pool in the presence of the hypothesized target compounds individually. We tested 255 compounds individually at 10 μM concentration. This second phase of analysis confirmed that at least 217 small molecules had a biosensor with a minimum of 2-fold change of cleavage. These molecules elicited an average fold change of cleavage of 4.2 in their corresponding biosensors. The maximum fold change of cleavage observed was 17-fold for compound 127E09 with biosensor 565359918. The analysis further identified 150 clusters of biosensor sequences, where each cluster exhibited a statistically different pattern of response to the compounds (Fig 4 and S5 Table). Note that the number of clusters is lower than the number of compounds due to the existence of groups of compounds that elicit similar responses from all the biosensors tested.

## DRIVER-selected biosensors span a wide range of sensitivities

We further measured the sensitivity of the 168 high-interest biosensors in the synthesized pool to each of the 14 target compounds that ranked highest in terms of the maximum fold change of cleavage they induced. The CleaveSeq assay was performed to measure cleavage of each sequence in the set of 168 high-interest biosensors in the presence of each of these 14 target compounds in a two-fold dilution series down to concentrations that did not produce a fold-change of cleavage of two or more (Fig 5). The data indicate that the minimum concentration of a target compound needed to elicit a two-fold change in cleavage varies from less than 25 nM to more than 5 μM. For some of the target compounds (247E06, 405D09, 247C07, 8G11) all characterized biosensors show similar responses and sensitivity, quantified by the average standard deviation of fold-change in cleavage at each concentration being less than 0.5. For the remaining compounds that effected a fold change of cleavage in multiple biosensors, different biosensor sequences exhibited different sensitivities. For example, 125F11 elicits a two-fold change of cleavage at 25 nM for biosensor 565770089, but for biosensor 565359918 requires up to 1 μM of the target compound. Furthermore, these biosensors show no sequence similarity in their stem loops and are responsive to distinct sets of compounds (S5 Table). We predicted the secondary structures for the sequences that sensed compound 125F11 using Vienna RNA-Fold [23] (Fig 6). It is interesting to note that despite the loops being randomly generated, the predicted secondary structures for biosensors against 125F11 share similarities with previously described aptamers. Experimentally derived structures for the theophylline, neomycin, and tetracycline aptamers consist of helices interrupted by an unpaired region, where the small molecule binds [24–26]. All of the biosensors against 125F11 share this motif in their loops. The range of fold-change of cleavage observed may be due to specifics of each biosensors tertiary structure leading to differing binding and cleavage dynamics. Taken together, the data indicate that the DRIVER method can generate multiple biosensors that exhibit a range of sensitivities and that likely have different mechanisms of operation.

The DRIVER selection was performed at concentrations of the target compounds of at least 1 μM, resulting in little selective pressure to obtain biosensors that respond to their cognate ligand at concentrations below that. We expect that conducting additional selections with the

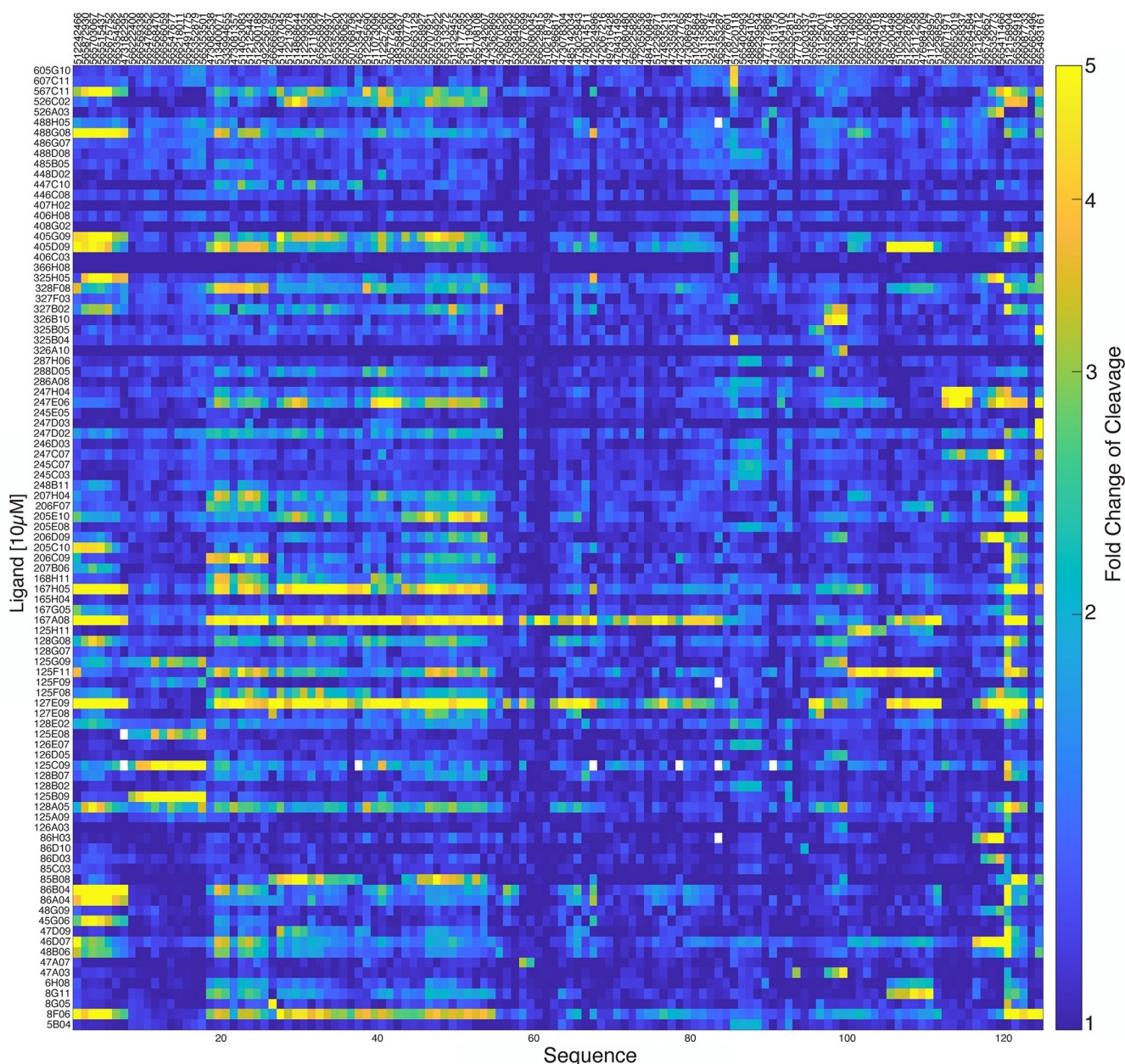

**Fig 4. CleaveSeq results of biosensors against individual compounds shows patterns of promiscuous and selective biosensors.** The pseudocolor plot shows the fold change of cleavage for sensors in the presence of single compounds at 10μM concentration. Sequences (after clustering as described in text) or compounds that result in at least 2-fold change in cleavage for at least one combination are shown.

enriched biosensor pools at lower target small molecule concentrations could be used to further enrich for higher-sensitivity biosensors.

To understand whether we were creating selective biosensors we considered the fold-change of cleavage of biosensors which sensed two or more structurally similar ligands (Fig 7). Compounds 325H05 and 325B05 share a common pyrido[1,2a]-pyrimidine core, each with a carboxamide bearing a pendant cyclic group (pyridyl and chlorobenzyl, respectively, highlighted in Fig 7a). Despite the common core, multiple biosensors distinguished between the two compounds. Biosensor 565675752 exhibited a 10 fold-change of cleavage in response

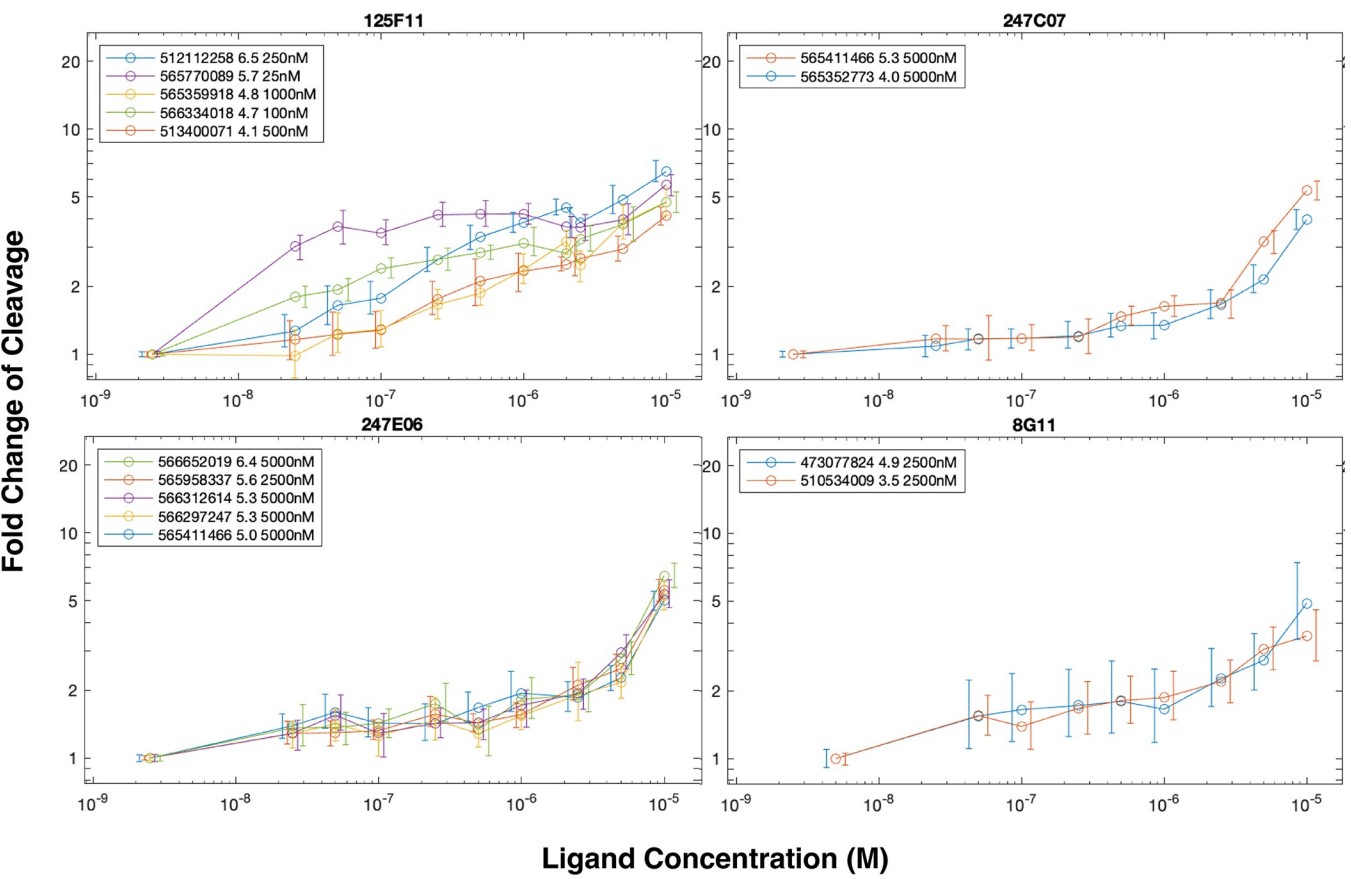

**Fig 5. Biosensors demonstrate a range of sensitives.** Fold change as a function of compound concentration is shown for selected aptamer-compound combinations that exhibited at least 3.5-fold change in cleavage at 10μM small molecule concentration. Error bars are 95% confidence intervals based on NGS read counts for single measurements at each concentration. Some measurements were made in parallel by combining up to 4 compounds in the same well, in which case only sequences that were not affected by the other compounds present are shown (based on single-target measurements at 10μM). Legend entries show the sequence IDs, fold change of cleavage at 10μM, and the minimum concentration measured that produces at least 2-fold change of cleavage.

to 325H05 vs 1.4 fold-change of cleavage in response to 325B05, while biosensor 565493161 had a higher fold-change of cleavage in response to 325B05 vs 325H05 (5.1 and 2.8, respectively).

A similar pattern holds for compounds 405D09 and 405G09, which share a common methyl-triazolo-phthalazine bearing pendant cyclic groups (furan and pyridine respectively, highlighted in Fig 7B). And for compounds 45G06, 86A04, and 86B04, all of which share a common chloro–8-methyl–4-methylamino quinoline with an ethyl carboxylate. Attached to the methylamino are a furan, ethanol, and morpholine, respectively (highlighted in Fig 7C). Finding multiple biosensors that can discriminate between two similar compounds supports the ability of this workflow to develop selective biosensors.

## Selection analytics show enrichment profiles of biosensors and amplicons

We retrospectively examined the selection path of sequences that were responsive to at least one target compound by measuring their relative abundance at least every four rounds during DRIVER selection using their NGS read counts (Fig 8). The analysis indicates that different biosensors arose at different points in the DRIVER selection process. Some sequences (e.g. 512112258 and 51340007) that were enriched early in the selection process were de-enriched

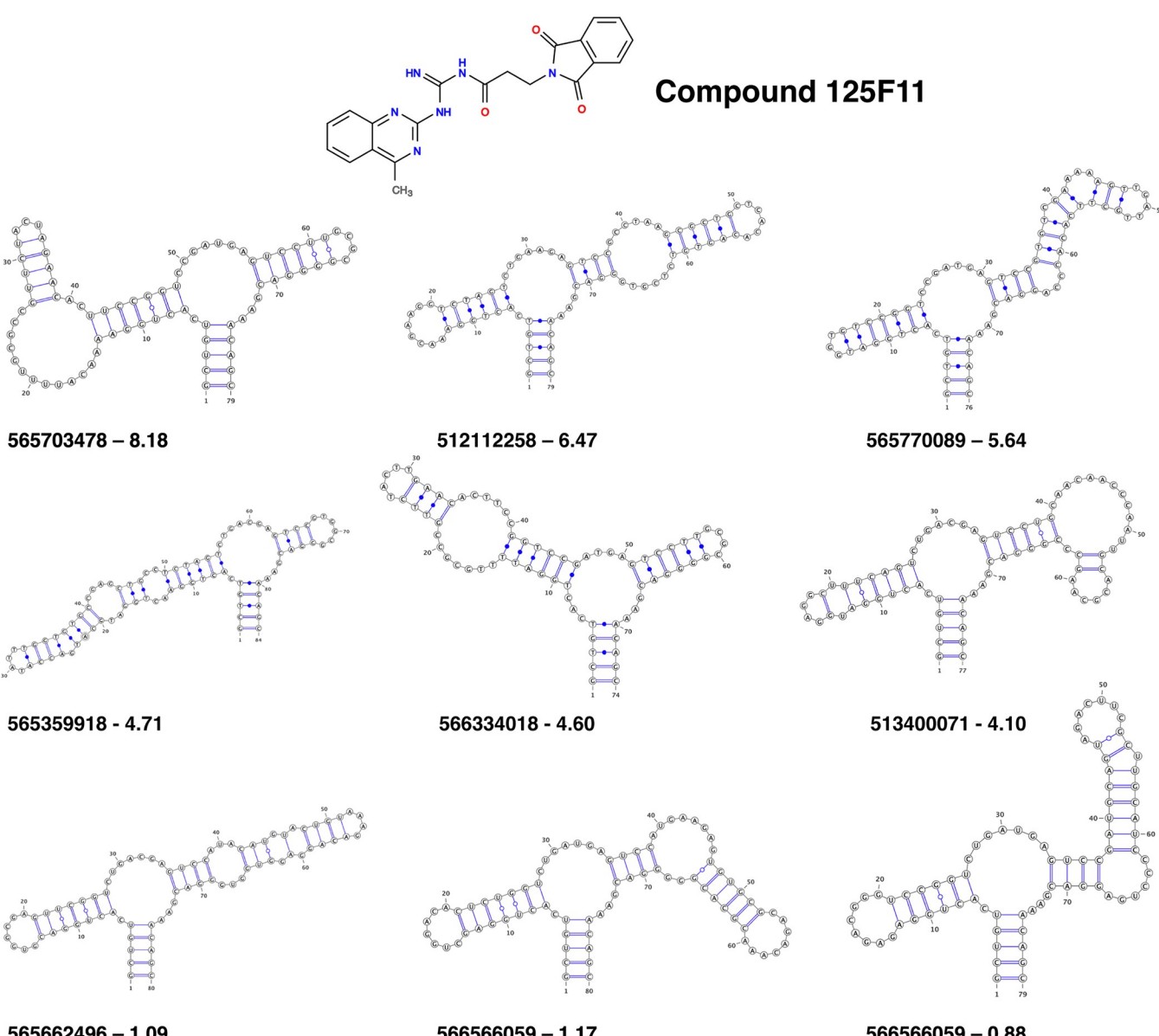

**Fig 6. Biosensor loops share a bulged-stem motif.** Secondary structures, predicted with RNAFold [23], for a subset of biosensors tested against compound 125F11. Biosensors identifier and the fold- change of cleavage at 10 μM are reported underneath the structures. Top two rows of biosensors had a >4 fold change of cleavage while the bottom row are examples of biosensors that had fold change of cleavage ratios of about 1.

at later rounds, likely due to competition from sequences with higher fitness. Also, some sequences (e.g. 565515437 and 565352773) were notably de-enriched between rounds 87 and 95, likely due to the negative selection pressure against the alternative V2560 compound mixture added in those rounds. Fitness during selection depends not only on the fold-change of cleavage exhibited by a sequence, but also the absolute cleavage levels at each condition. Sequences with fraction cleaved centered around 50% have higher fitness than those with very high or very low fractions cleaved, as only sequences that cleave during negative selection rounds and do not cleave during positive selection rounds will survive the selection.

Undesired amplicons remained at low levels throughout the DRIVER selection process (Fig 8), but note that the enriched biosensor libraries contained many sequences with an embedded

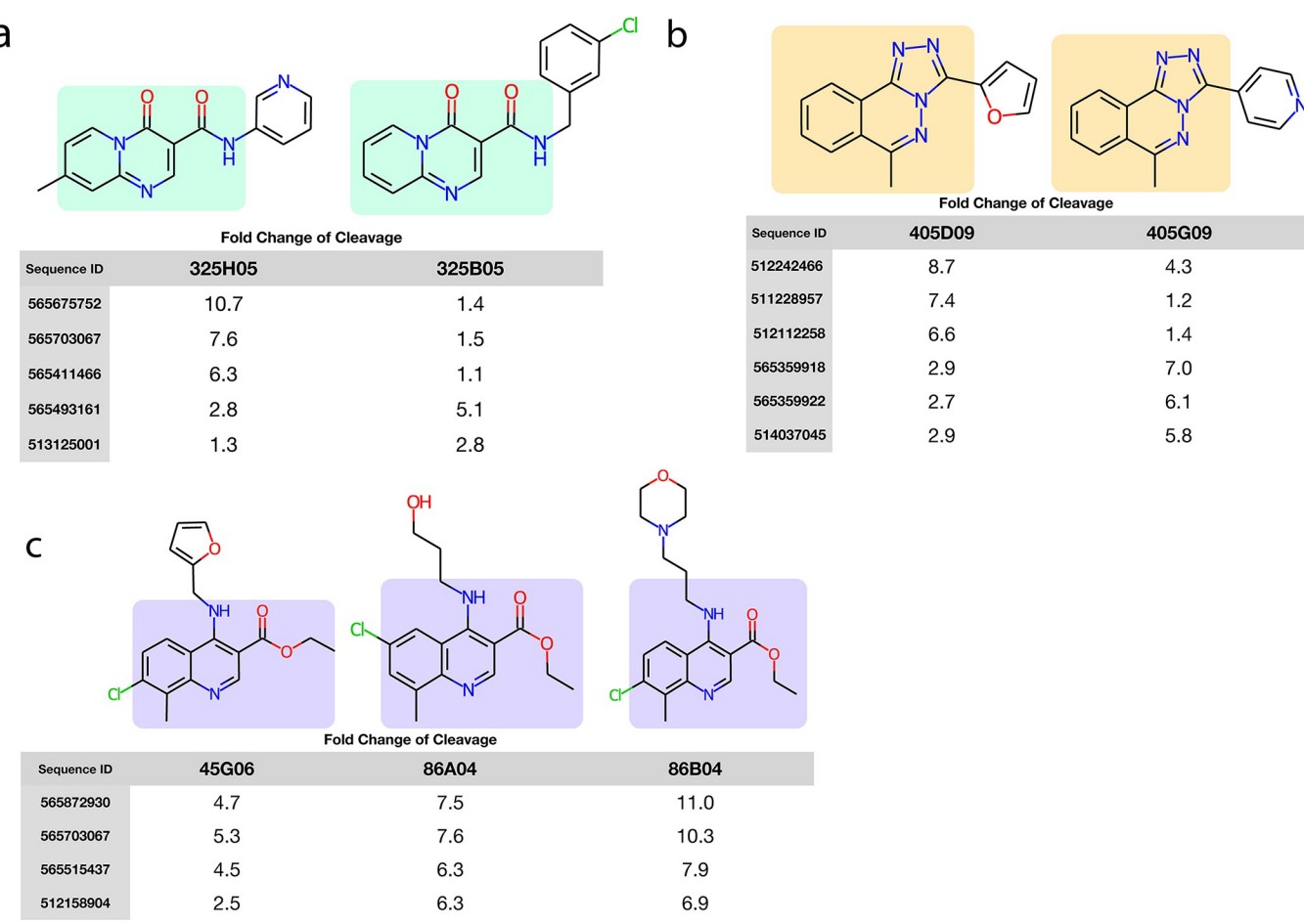

**Fig 7. Biosensors can discriminate between similar compounds that differ by a single functional group.** Each panel consists of a set of similar compounds and a selection of biosensors that show selective sensing between the compounds. Measurements were taken individually with all compounds at the same concentrations. **(a)** Compounds 325H05 and 325B05 share a common pyrido[1,2a]pyrimidine core each with a carboxamide bearing a pendant cyclic group (pyridyl and chlorobenzyl, respectively). **(b)** Compounds 405D09 and 405G09, share a common methyl-triazolo-phthalazine bearing pendant cyclic groups, furan and pyridine, respectively. **(c)** Compounds 45G06, 86A04 and 86B04, all of which share a common chloro–8-methyl–4-methylamino quinoline with an ethyl carboxylate. Attached to the methylamino are a furan, ethanol, and morpholine, respectively.

region similar to the last several nucleotides of the ribozyme. These sequences roughly correspond to the nucleotides that pair with the reverse transcription (RT) primer (S1 Fig). We postulate that these sequences enable the RNA to fold into a ribozyme-active conformation without using the region that pairs with the RT primer. The RT primer was designed to bind to parts of stems II and III of the ribozyme to inhibit its catalytic activity prior to increasing the concentration of $Mg^{2+}$, which is needed for the reverse transcription step. The sequences which evade this inhibition can cleave during this RT step, likely giving them a fitness advantage in the selection process. Although we isolated functional biosensors with and without this embedded sequence, the properties of these biosensors may differ, e.g., in terms of their $Mg^{2+}$ dependence, and the impact of this mechanism may require further study.

## DRIVER-selected biosensors exhibit a wide range of selectivities

The DRIVER selection strategy employed in this study was designed to efficiently identify as many biosensors as possible from the RNA biosensor library. Other than the final 8 selection rounds, enrichment did not depend upon selectivity of the aptamer sequences to particular

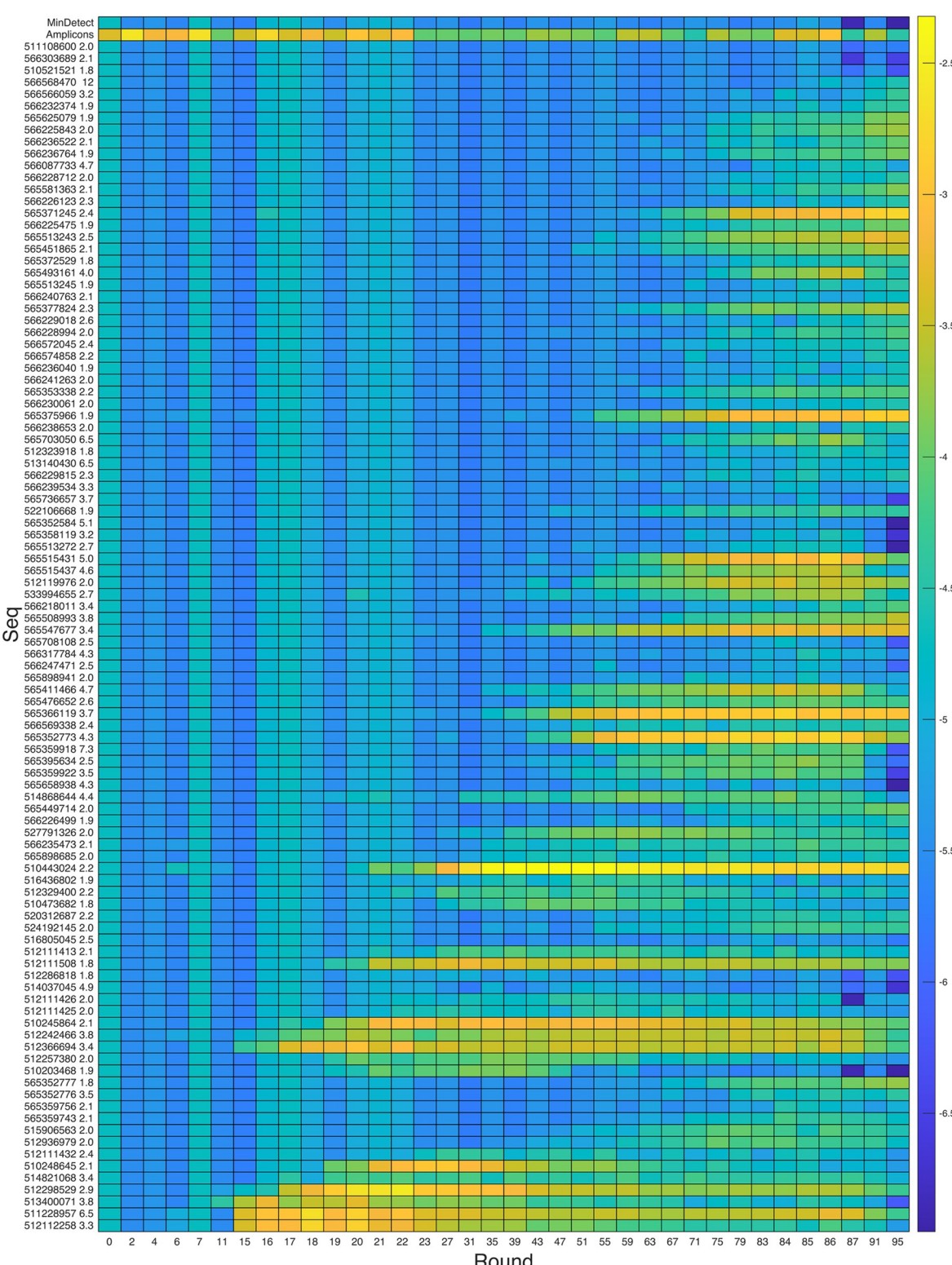

**Fig 8. Tracking biosensors enrichment over selection rounds reveals amplicons make up large portion of pool early in selection before being outcompeted by true biosensors.** Relative abundance is shown over the course of the selection as measured by sequencing of the products of the indicated rounds. The top row shows the minimum detectable abundance based on the total number of sequencing reads for each round, and the second row shows the abundance of short amplicons. The remaining rows show the 100 biosensors with the highest fold-change of those detected at round 87 or 95. Sequence ID and fold change of cleavage are shown along the y-axis labels and the pseudocolor represents $\log_{10}$(abundance).

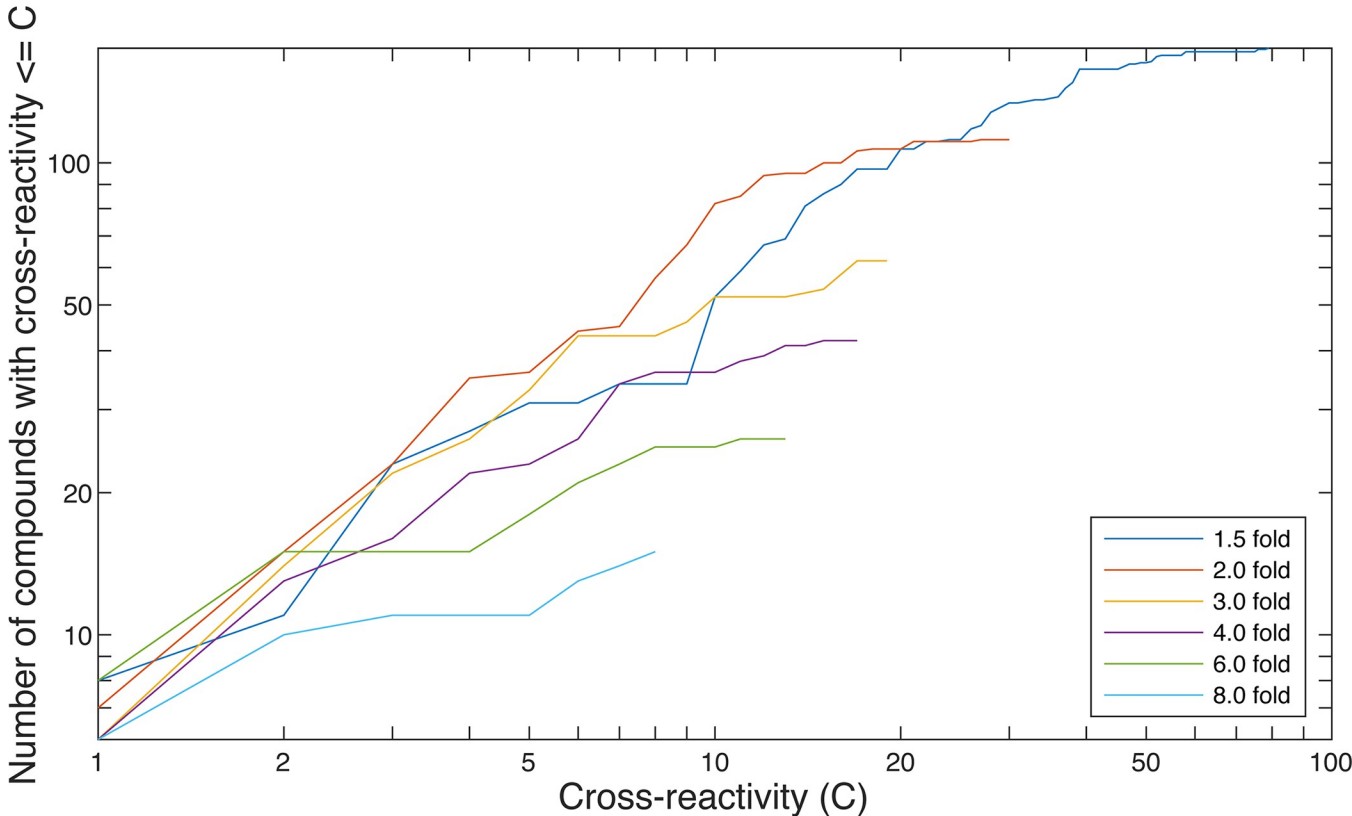

**Fig 9. Selective biosensors tend to be more sensitive.** At a given fold-change of cleavage, *f*, the cross-reactivity of a sensor is defined as the number of compounds that induce fold-change of at least *f*. The cross-reactivity for a compound is then defined as the lowest cross-reactivity of all sensors that respond to the compound with fold-change of at least *f*. The number of compounds with cross-reactivity less than C is shown as a function of C. For example, at $C = 1$, the lines indicate the number of compounds that uniquely induce the indicated fold-change in some sensor. As the fold-change of cleavage increases, fewer molecules cross-react, indicating that more sensitive biosensors may be more selective.

target ligands. As a result of the designed selection strategy, the identified biosensors span a wide range of selectivity, from biosensors that are sensitive to only a single compound within the target compound library of 5,120 to those that respond to at least 100 compounds within the library (Fig 9). Although it was not a goal of this study, we expect that biosensors with low cross-reactivity can be enriched by appropriate choice of conditions during the negative counterselection rounds, such as by inclusion of compounds for which low cross-reactivity is desired.

Our initial hypothesis was that target compounds with similar structure would elicit a response in the same biosensors resulting in low selectivity between these target compounds. For each identified biosensor sequence, the target compounds in the library to which the sequence was determined to be responsive were compared to identify any similarity in structure that may indicate a shared substructure that the biosensor specifically recognizes. The chemical structures are shown in Fig 10 and S1 File. In a few cases there is a shared substructure between the target compounds that is readily evident. For example, from the data in S5 Table, biosensor sequence 565476652 has a fold change of cleavage of 5.2 and 3.9, respectively, when transcribed in the presence (at 10 µM) of small molecules 125B09 and 125C09, which differ only in site of attachment of the flanking pyridine rings. Biosensor sequence 565366119 and several others are similarly affected by these two compounds (S1 File). Biosensor sequence 565958337 exhibits a fold change of cleavage of 6.8, 5.4, and 3.3 when exposed to 247H04, 247E06, and 247C07 (at 10 µM), all of which share a common central substructure.

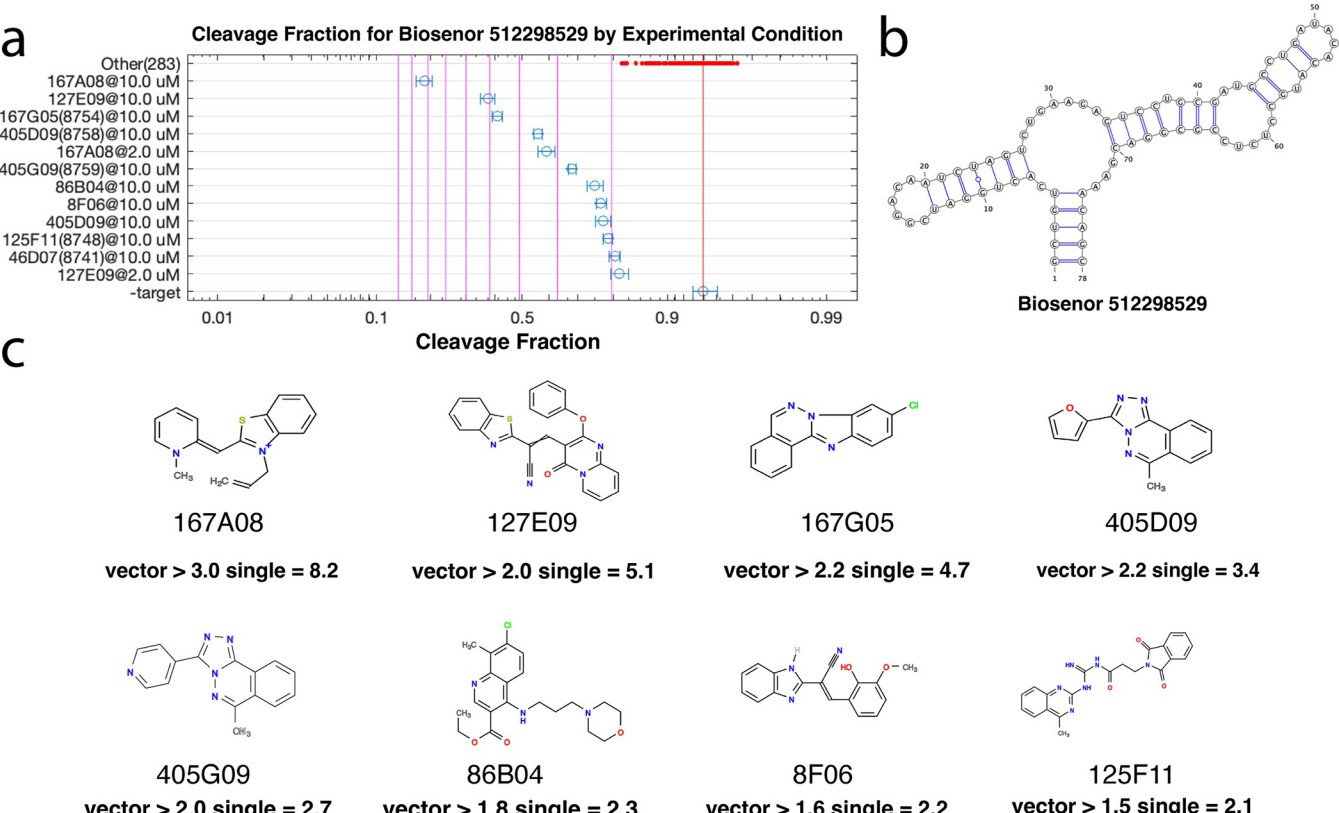

**Fig 10. Promiscuous biosensors can sense multiple diverse ligands.** For each cluster of sequences that have a similar response to the compounds, a representative sequence was chosen and shown along with its sequence (one particular sequence, 512298529, is shown above and the others in S1 File). **(a)** A chart of the cleavage of that sequence in the presence of each compound is then shown. Compounds that may give rise to a fold change that exceeds 2-fold (upper CI bound $>$ = 2) are shown explicitly and all others that were individually measured are shown by the red points at the top of the chart. Error bars indicate the 95% confidence interval based on read count statistics. Vertical lines indicate the fold change of cleavage from the no ligand condition. **(b)** Predicted secondary structure of 512298529. **(c)** The chemical structure of each compound with fold change$>$ = 2.0 is also shown. The measured fold change of cleavage are listed below each compound, as derived from either single-compound measurements ("single") and as an lower-bound estimate o from the measurement of fold change of cleavage from the 256-compound vectors ("vector").

To quantify the degree to which shared substructures explain the cross-reactivity of the biosensors, we built a predictor of fold-change of cleavage from the fragments present in each compound. We used the fold-change of cleavage data collected for single compounds to establish each [biosensor, compound] pair as either a hit (lower bound of the 90% confidence interval for the fold change $> 2.0$), a miss (upper bound of the 90% confidence interval $< 2.0$), or indeterminate. We then trained random forest classifiers to predict each hit or miss using all of the other hit/miss data for that biosensor via leave-one-out cross-validation. The input to the classifiers was a bit vector for each compound, other than the one being predicted, indicating the fragments from a 23,595-entry fragment library that were present in the compound's structure. The random forests were trained using the hit/miss data for each compound, and then used to predict the classification of each compound. This method was applied to the 107 biosensors with three or more hits of the 150 biosensors that showed distinct patterns of activation (biosensors with two or fewer hits cannot be modeled in this way due to lack of training data). We then compared the predictions to the hit/miss measurements and tabulated counts of false and true positives and negatives for each biosensor (S6 Table). Over this entire population of biosensors, we observed 25,162 true-negatives, 708 true-positives, 154 false-positives, 602 false-negatives, giving an area under curve (AUC) of 0.77 and a precision of 0.82. That is,

in 82% of the cases that the classifier indicated a hit, the compound was indeed a hit. This experiment establishes a lower bound on the ability to predict the cross-reactivity of the selected biosensors and their response to other compounds—future experiments that focus on exploring other machine learning models and methods or the use of other feature sets may obtain higher performance.

## Conclusions

Through this work, we have demonstrated the flexibility of DRIVER to select for multiple small molecule compounds in parallel. We have generated RNA biosensors that can detect 217 new small molecules, more than doubling the number of small molecules that can be sensed by a known nucleic acid aptamer [13]. The small-molecule compound library was designed for drug screening and its members are structurally similar to many natural and synthetic molecules of interest. The RNA biosensors we identified can each sense one or more small molecules with 2-fold to 14-fold change of cleavage activities and exhibit high nanomolar to millimolar sensitivities. We also implemented various improvements to the DRIVER protocol including negative-selection protocols, rapid amplicon identification and mitigation, and rapid identification of small molecule targets in a large mixture through vector-based Cleave-Seq. Additionally, we developed methods to handle mixing and deconvolving large small molecule compound libraries using liquid handling robots and LC-QTOF/MS for validation.

Earlier work demonstrated that biosensors generated via DRIVER can function as selective *in-vivo* sensors of small molecule concentration [17]. The work described here provides a proof of concept for selecting hundreds of small-molecule biosensors at once using DRIVER. Using this approach it may, for example, be possible to efficiently create RNA biosensors against all members of a metabolic pathway for real-time tracking of natural-product production [27]. Although the metabolites in a metabolic pathway are chemically similar, we envision the results of a DRIVER selection serving as the starting point to fine-tune distinct sequences that sense and distinguish between similar metabolites. After initial selection, counter-selection, with all-but-one mixes of ligands of interest can be performed to generate highly selective biosensors. The current CleaveSeq detection protocol paired with RNA biosensors supports rapid detection of specific small molecules in a mixed pool. The CleaveSeq reaction occurs in a few minutes and can be read out through sequencing or detection of RNA fragments [19]. Previous work has demonstrated that selections can be performed *in vivo*, however these methods are limited in their throughput due to cell density constraints. A possible application of DRIVER is to use DRIVER output as a starting point for subsequent rounds of *in vivo* selection to optimize the performance of the biosensors in the desired host environment [17, 18, 21]. In this study we demonstrated the feasibility of utilizing DRIVER-selected RNA biosensors in mixtures with multiple small molecules, demonstrating their ability to be highly selective in a heterogenous population.

In addition, the ability for one RNA biosensor sequence to sense multiple different molecular structures (Fig 10) may allow for the creation of an RNA based in-solution electronic nose device in future work [28, 29]. Such devices utilize detectors that are sensitive to multiple small-molecule features at differing levels. By carefully measuring and calibrating sensitivities to known mixtures of small molecules it may be possible to quickly identify and deconvolve a new mixture.

In this study we successfully measured and deconvolved sensors against 217 small molecules. A fundamental question is what limited the number of sensors generated given the large size of our small molecule compound library. We consider three main possibilities for this: that only ~4% of the molecules in the library are amenable to sensing by an RNA aptamer; that the parallel selection process employed results in a subset of the possible sensors masking

other sensors that may be enriched more slowly; or that if we had continued running DRIVER or modified the selection conditions, we would have continued to find new biosensors. Thorough systematic examination in future work will be directed to resolving this question.

One main objective of this work was to apply the DRIVER approach to generate many biosensors in single selection experiments. The selection methods were not optimized for obtaining highly selective aptamers, including aptamers against one unique target small molecule, or refined to increase sensitivity. We expect that the sensitivity of the resulting biosensors could be increased by systematically lowering the concentration of the target small molecules in later rounds of selection. We also expect that the selectivities of the resulting biosensors can be tailored with the addition of distinct set(s) of small molecules to the negative selective rounds. For example, if the biosensors are ultimately intended for use in yeast cell applications, negative selection rounds can be performed to compound mixtures containing small molecules commonly found in yeast cytoplasm. These subsequent selection rounds would focus on removing any biosensors which are also sensitive to cellular ligands, thus making them more selective. Another situation where negative selection rounds may be used is to select against biosensors which respond to commonly found small molecule backbones. When selections are performed against large ligand libraries there will be shared chemical substructures between ligands, which can effectively increase the concentration of the substructure relative to individual full molecules. Future studies may explore if using common substructures during negative selection rounds would enable the selection of biosensors that are more sensitive to less common substructures, or even the *entire* ligand, rather than a substructure. In this study we explored both negative selection and lowering the concentration of ligands during selection. However, further work is needed to systematically test and evaluate conditions for sensor enrichment in complex target compound mixtures.

Future work may also be directed to investigating conditions that raise the total number of new molecular sensors that are enriched. Modified conditions may include lowering the number of molecules in a compound selection mixture and increasing the total number of rounds that the selection is run for. Various factors may influence the enrichment efficiency. For example, it is possible that some sequences in the library detect functional groups that are shared between different small molecules, such that these functional groups are at higher concentrations in the mixtures than any individual molecule, resulting in faster enrichment of these biosensors. This situation may result in the generation of biosensors that outcompete highly-selective biosensors due to the higher concentration of shared functional groups in the mixture. A better understanding of these factors will allow for the design of more effective DRIVER selection protocols for small molecule biosensors.

Our work provides rich data sets (see additional data section) of the activities for many thousands of RNA-ligand combinations. These data can be used not only to gain a deeper understanding of directed selection experiments but also to train computational models. As demonstrated in this work, models can be built to predict the activity of the selected biosensors to new compounds allowing the biosensors to be used for other compounds outside the library used for selection. More generally, the large data sets generated in this study provide an opportunity for further analyses to gain deeper understanding of RNA-ligand interactions and can be used to train and test computational predictors of these interactions and/or of RNA structure [30].

## Methods and materials

### Compound library

Compounds used in this work were obtained (ChemDiv, San Diego, CA) as part of a custom diversity library and subsequently reformatted by Stanford High-Throughput Biosciences

Center (HTBC) into 384-well plates containing each compound at 5 mM in DMSO. Sixteen of these plates, uniformly spaced (each 10th plate) from the full set were chosen to reduce any systematic bias. These plates were then diluted with 95 μl of MeOH to make the volumes more manageable, the various selection and validation mixtures were created on an automated liquid handler (Tecan Freedom Evo), and then the MeOH was evaporated by leaving the plates uncovered in a fume hood overnight (Fig 1C). The selection mixtures were further concentrated by evaporation on a rotary evaporator (Buchi rotovap connected to Edwards RV8 vacuum pump) at ~0 millibar for 6–8 hours until the concentration exceeded 20 μM. Concentration was then adjusted to 20 μM by addition of DMSO.

Compounds chosen for source validation (S1 Table) were purchased (Chem-Space, New Jersey) and suspended at 10 mM by addition of DMSO to 1 mg of compound. These were then diluted as needed for use in CleaveSeq or QTOF analysis.

## Sensor library

The initial library was synthesized by IDT (Integrated DNA Technologies), as 10 separate oligonucleotides, one for each particular set of lengths of the two ribozyme loops (S2 Table). Note that the library was designed slightly differently from previously described [17] in that the stem I helix sequence was changed from ACCGGA:TCCGGT to ACTGGA:TCCGGT. This modification changes one base-pair in the RNA helix from GC to GU, but otherwise leaves the RNA structure unchanged. However, the change destabilizes this helix in the single-stranded cDNA following reverse transcription, improving the ability of the splint oligonucleotides to hybridize with the cDNA. Also, only N30 aptamer loops were used.

Each oligo used hand-mixed degenerate bases (25:25:25:25) for the loops and were PAGE-purified by IDT. The oligos were suspended in duplex buffer (30 mM HEPES, pH 7.5; 100 mM potassium acetate) at 100 μM and then 2 μl of each was mixed along with 33 μl of the complementary T7 promoter at 100 μM (T7p, S2 Table) and an additional 24.7 μl of duplex buffer. This mixture was heated to 95°C for 5 minutes, cooled to 58°C at 0.1°C/s, held at 58°C for 5 minutes and then cooled to 25°C at 0.1°C/s. This 30 μM mixture was then used as input to the first round of selection. The library was sequenced on an Illumina iSeq sequencer to verify the composition and statistics.

## DRIVER selection

The DRIVER method was adapted [17] was modified from previous work. Major adjustments from the previously published method are outlined in this paragraph and the full method is described in the rest of the section. The oligonucleotide used for the reverse transcription priming and ligation of the cDNA products was slightly modified from the previous method to improve ligation efficiency with the modified stem I sequence described above (Z_Splint, W_Splint; S2 Table) In addition, in the original DRIVER method a different reverse transcription primer was used in negative selection rounds since no ligation was needed. However, this may result in enrichment of sequences that anneal differentially to the different reverse transcription primer sequences, allowing these sequences to escape the desired selection pressure. In this work, the same splint oligonucleotide was used for the reverse transcription for both negative and positive selection rounds, though remained dependent on the prefix of the template coming into the round.

The first round of selection used the sensor library described above, at a final concentration of 400 nM, in two separate 1 ml transcription reactions. Each transcription reaction consisted of 9 mM rNTPs (NEB N0466), 10 mM Dithiothreitol (Invitrogen), and 5 U/μl T7 RNA polymerase (NEB M0251) in 1x RNAPol buffer (NEB). The transcription reactions were incubated

for 145 minutes at 37˚C in a thermocycler and were then combined, mixed and part was immediately used in the next step, with the remainder stored at -80˚C. The concentration of the transcription was measured as 7 μM using a Qubit RNA assay (ThermoFisher). A splint oligonucleotide (Z_Splint. S2 Table, 72 μl at 10 μM) was then added to 103 μl of the transcription reaction and mixed well. Based on the RNA gain of the transcription reaction and the Poisson sampling statistics, the diversity of the library at this step was approximately $10^{14}$. A reverse transcription master mix was then mixed using 36 μl of Omniscript buffer at 10x, 36 μl of dNTPs at 5 mM, 57.6 μl of $MgCl_2$ at 25 mM, 18 μl of Omniscript enzyme at 4 U/μl (Omniscript RT Kit, Qiagen), and 37 μl water. The master mix was added to the primed transcription mix, mixed well, split into 6 tubes containing 60 μl each, and incubated for 60 minutes at 50˚C, followed by heat inactivation at 95˚C for 2 minutes. The tubes were combined and all but 5 μl was immediately used in the following step. For the ligation step, 439 μl of water, 89 μl of 10x T4 DNA Ligase Buffer (NEB, B0202), and 4.4 μl of T4 DNA Ligase at 400 U/μl (NEB, M0202) were added to the reaction and incubated for 30 minutes at 37˚C followed by heat inactivation at 65˚C for 10 minutes. All but 5 μl of this product was then diluted 20x into a PCR reaction that consisted of 1x Taq buffer (NEB, B9014), 1 mM $MgCl_2$, 200 μM dNTPs (Kapa, KK1017), Hot-Start Taq (NEB, M0495), 0.01 U/μl USER enzyme (NEB, M5505), 300 nM primers (T7Z and X, S2 Table), and 2 μM blocking oligo (WBlock, S2 Table). The mixture was incubated at 37˚C for 15 minutes (for USER digestion) and then the following program was run in a thermocycler: 95˚C for 30 seconds followed by 9 cycles of (95˚C for 30 seconds, 57˚C for 30 seconds, and 68˚C for 30 seconds) with a final extension of 68˚C for 60 seconds. The resulting product was purified using 4 spin columns (Zymo, DCC-25) to produce the round 1 product.

The above method was repeated for six additional rounds of selection, alternating between the Z_Splint and W_Splint RT primers and between the T7Z and T7W PCR primers since the prefix of the product of each round alternates between W and Z. During these rounds, the volumes of the reactions were decreased during the T7 transcription to 944, 750, 372, 250, and 125 μl during rounds 2 through 6, respectively. This procedure was based on the computed diversity of the products such that at least 50% of the sequences present in round 1 that exhibit 70% cleavage should still be present in the library at round 6.

Starting with the product from round 7, two parallel selections, A and B, were run with V2560A added during the transcription steps in the A selection and V2560B in the B selection, in each case at 2 μM (total of all compounds) final. Rounds 7 and 8 selected for non-cleavers in the presence of the compounds and subsequent rounds alternated between cleavage selection in the absence of compounds and non-cleaver selection in the presence of the compounds. Starting with round 88, the compound concentration was reduced to 1 μM based on the hypothesis that this would help increase biosensor sensitivity by creating a steeper fitness landscape. Starting with round 88, the alternate compound mixture (i.e. B for the A selections and vice versa) was added to the transcription reactions during the negative selection rounds, which we hypothesized would help increase selectivity by removing biosensors that responded to compounds in both the A and B groups. Rounds 8 through 95 were implemented on a liquid handler. Further details of the parameters of each round are shown in S3 Table.

## Resynthesis of biosensors

Specific biosensors identified during the selection and subsequent CleaveSeq analysis were resynthesized on an oligonucleotide array (Agilent, G7220A). The array contained 1,730 sequences each padded to a length of 158 nt. These consisted of the desired biosensors prefixed and suffixed with additional sequence (W_Prefix, X_Suffix; S2 Table), and then surrounded by one of nine different pairs of 24-nt primer sites to allow selective PCR amplification of specific

parts of the library. The library was PCR amplified using the corresponding PCR primers to form nine sublibraries. These were further PCR amplified using the T7W and X primers to remove the other priming sites and add the T7 promoter prefix. The design of each sublibrary and the sequences it contains are provided in S4 Table.

## CleaveSeq

Each CleaveSeq reaction begins with T7 transcription of the library to be tested: 20–100 nM template, 1× RNApol buffer, 9 mM ribonucleoside tri-phosphates (rNTPs), 5 U/μl T7 RNA polymerase (New England Biolabs), 1 U/μl SUPERase In (Thermo Fisher Scientific), and 10 mM dithiothreitol (DTT). The excess rNTPs over standard T7 polymerase conditions result in chelation of most of the free $Mg^{2+}$, providing a rough approximation to sub-millimolar cellular $Mg^{2+}$ concentrations, thereby making the selection conditions more representative of *in vivo* cellular conditions and reducing the rate of ribozyme cleavage. The transcription reactions were incubated at 37°C for 15–30 min, during which time the transcribed RNA may undergo self-cleavage depending on the catalytic activity of the particular library sequence. The RNA products from the transcription reaction were immediately transformed to cDNA in a RT reaction. The RNA products were diluted 2× and mixed with a reverse primer at 2 μM final. Annealing of the RT primer to the RNA partially unfolds the ribozyme, thereby stopping the cleavage reaction. For uncleaved selection rounds, the RT primer consisted of the reverse complement of the expected RNA sequence from the 3′ leg of the stem II helix through the "X" spacer. For cleaved selection rounds, the RT primer was prepended with an additional sequence to assist in the subsequent ligation step (S2 Table; BT1316p for rounds that started with a "Z" prefix, BT1508p for those with a "W" prefix). This mixture was diluted a further 2× into an Omniscript (Qiagen) RT reaction following the manufacturer's instructions and incubated at 50°C for 20 min followed by heat inactivation at 95°C for 2 min. The reaction products were then slow-cooled to 25°C at 0.5°C/s to allow refolding of the cDNA.

The reaction was split and run through two separate PCR reactions, one that amplified the cleaved components with the same splint/reverse transcription oligonucleotide as was used for selection of "W"-prefixed rounds. The other reaction amplified the uncleaved components with a "W" prefix. The primers used in the above PCR reactions included 5′-overhang regions with Illumina adapters and barcodes to allow each read to be identified as to the assay conditions. In addition to the standard Illumina index barcodes embedded in the adapters, we also added 1–10 nucleotides of custom barcode nucleotides between the Illumina adapters and the prefixes or suffixes (S2 Table; "NGS Primer"). The variable length barcodes introduce shifts of otherwise identical sequence positions in the prefix and suffix regions of the DNA being sequenced, resulting in more equal distribution of the four nucleotides at each position. This strategy improves the performance of Illumina sequencers' clustering step, which relies on distinct sequences in adjacent clusters during the first several sequencing cycles. During the analysis, the number of reads of reference sequences provides a conversion factor for equating the number of reads with absolute concentration. The PCR reaction mixtures (1× Kapa HiFi enzyme, 1× Kapa HiFi buffer, 400 nM primers) were run for 18 cycles (under the following conditions: 98°C for 30 s, 57°C for 30 s, and 72°C for 30 s).

The barcoded libraries were mixed in ratios based on the relative number of reads desired for each library and the libraries were diluted to 4 nM of DNA with Illumina adapters as quantified by qPCR (KAPA Library Quantification Kit). PhiX was spiked into the sequencing library at 10–20% of the total library concentration to further improve the cluster calling of the Illumina pipeline for amplicons. The libraries were sequenced on an Illumina platform, either MiSeq (using MiSeq Control software v3.0) or NextSeq (using NextSeq Control software

v2.1.0) using 2×75 or 2×150 reads, depending on the data needs of a particular experiment, in each case using Illumina recommended loading guidelines.

All of the CleaveSeq runs were performed on a liquid handler on up to 48 samples in parallel using the same parameters for all runs, with only the choice of input template library and addition of compounds varying. The template library under test, either from a selection round product or synthesized set of oligonucleotides, were diluted to 1 nM in the transcription reaction to reduce the carry-forward of templates into the sequencing results. Compounds or mixtures of compounds were added to the starting wells using 10x stock in 100% DMSO, resulting in 10x dilution into the aqueous transcription buffer. DMSO alone (with a 10x dilution) was used in reactions that did not have any compounds added.

## Next generation sequencing of DRIVER rounds for biosensor analysis

Ligation products from the CleaveSeq reactions were diluted 25x in TE8 (10 mM Tris, 0.1 mM EDTA, pH 8) to stop the reaction. The circular DNA resulting from the ligation reaction was then cut and the splint region excised. This reaction consisted of 0.05U/μl Uracil-DNA Glycosylase (NEB; M0280), 0.1 U/μl Endonuclease IV (NEB; M0304), 1x ThermoPol buffer (NEB; B9004), and 2 μl of a diluted CleaveSeq reaction in a total volume of 10 μl. The reactions were incubated for 15 minutes at 37°C and then heat-inactivated at 85°C for 20 minutes. The advantage of this reaction over the USER treatment employed during selection is that the 3' end of the products of the UDG (or USER) reaction have a terminal phosphate that would block subsequent PCR extension. Since subsequent barcoding steps use 3'-blocked primers, the Endonuclease IV used here is necessary to dephosphorylate the 3' end.

The extension reaction is followed by a PCR reaction by addition of 4 μl of PCR1 master mix such that the reactions contain the diluted excision reaction, 1x ThermoPol buffer (NEB, B9014), 1 mM $MgCl_2$, 200 μM dNTPs (Kapa, KK1017), 2 ng/μl salmon sperm DNA (LifeTech; AM9680), 200 nM primers, and 1 U/μl HotStart Taq (NEB, M0495). The primers for this reaction were designed to overlap the prefix and suffix regions and extend them with Illumina read sequences. Half of the reactions use the primers WFU, ZFC, and XRC and the other half use WRU, ZRC, and XFC (S2 Table), where the two sets add the Illumina adapters in opposite orientations, improving diversity of the final library which in turn improves yield. All of these primers have their 3'-ends capped by addition of a 3-carbon spacer during oligo synthesis to ensure that all sequence reads resulted from the template sequence and were due to correction by the primers. These primers were synthesized by IDT and PAGE-purified. The PCR1 reaction was run on a thermocycler as follows: 95°C for 30 seconds followed by 5 cycles of (95°C for 30 seconds, 57°C for 30 seconds, 68°C for 30 seconds) with a final extension of 68°C for 60 seconds.

The PCR1 reaction was then diluted 10x by addition of water and used as input to a second PCR reaction to add multiplexing primers. This reaction consisted of 1 μl of the diluted PCR1 products, 5 μl of Kapa HiFi Fidelity Buffer, 0.75 μl of Kapa dNTP Mix at 10 mM, 0.5 μl of Kapa HiFi enzyme at 1U/μl (Roche, KK2103), and 1 μl of a dual unique index multiplex primer pair (NEB; E6440) in a total reaction volume of 25 μl. The PCR2 reaction was run on a thermocycler as follows: 95°C for 180 seconds followed by 14 cycles of (98°C for 30 seconds, 64°C for 30 seconds and 64°C for 30 seconds) with a final extension at 72°C for 60 seconds.

The PCR2 products were purified using a 1.8x SPRI cleanup (Omega Biotek; M1378) following the manufacturer's protocol. These were then quantified by qPCR using a KAPA Library Quantification Kit (Roche; KK4844) on a BioRad iCycler. Multiple products with distinct index sequences were then mixed in ratios depending on the relative read counts desired. Sequencing was performed on either an iSeq 100 or NextSeq 550.

## CleaveSeq analysis

Sequencing data was demultiplexed using the index codes and paired ends were assembled using PEAR [31]. Custom software was used to reduce these data to a list of the distinct sequences with total read counts for each. Prefix, suffix, and ribozyme regions were then identified and combined to give a count of reads for each distinct ribozyme with each prefix. Since the W prefix reads corresponded to uncleaved ribozymes and the Z prefix ones corresponded to the cleaved ribozymes, the ratio of these reads was used as an estimate of the cleaved: uncleaved fraction for each sequence. Fold change of cleavage was then computed as the ratio of these fractions over two conditions; typically a condition that included an added compound compared to one with no additions. Slight variations in ratios due to sequencing biases were corrected by use of reference sequences that were known to not be affected by the difference in conditions.

## Mass spectrometry

The compound library was analyzed by LC-MS using an Agilent 6545 Q-TOF mass spectrometer with Agilent 1290 Infinity II UHPLC (Stanford ChemH Metabolomics Knowledge Center). Chromatography was done on a ZorbaxRapid Resolution High Definition Column, 1.8 μm (Agilent) column with HPLC-grade (Thermo-Fisher) water with 0.1% Formic acid as solvent A and HPLC-grade acetonitrile with 0.1% formic acid (Thermo-Fisher) as solvent B. A volume of 10 μL of sample in DMSO were injected between 250 nM and 1 μM, and run at a constant rate of 0.4 mL per minute at 40˚C. Separation was performed with the following gradient: 0–18 min, 3–50% B; 18–27 min, 50–97% B; 27–30 min, 97% B; followed by a 5 minute equilibration at 3% B. LC Eluent was sent to the MS starting at 0 min. The MS was in Dual Agilent Jet Stream electrospray ionization (AJS ESI) in positive mode, source gas temperature at 300˚C, gas flow rate of 11 l/min, and nebulizer pressure of 35 psi. Data was collected using the MassHunter Workstation LC/MS Data Acquisition software (Agilent). Data files were converted into mzML format using MSConvert (Proteowizard).

Analysis was performed using Matlab, with code available at https://github.com/btownshend.

## Small molecule library composition confirmation through QTOF mass spectrometry

To spot-check that the selection and subsequent characterizations were not due to any contaminants that may have been present in the manufacturer's chemical library or due to subsequent handling, we ran the following control experiment. Small molecule compounds that produced at least 3-fold change of cleavage in our validation CleaveSeq runs in any of the tested RNA biosensors and were readily available from manufacturers other than the original source. Solutions were prepared from new stock and independently tested using CleaveSeq. We sourced 28 such compounds independently and ran CleaveSeq assays of the biosensor pool in the presence of each of these at 10 μM concentration and compared the observed fold-change of cleavage with those using the original preparations of the same compounds (S3 Fig). Of these, 26 showed similar fold-change of cleavage to the original measurements for sequences that elicited at least 2-fold change of cleavage, with two notable exceptions. CDIQ165-N09 showed higher cleavage fold-change in the presence of the second-sourced chemical by approximately 5x and CDIQ125-J17 showed lower cleavage fold-change by approximately 2.5x. Samples from both sources for each of CDIQ165-N09 and CDIQ125-J17 were analyzed with mass spectrometry. Neither preparation of CDIQ125-J17 had clear peaks

at expected m/z's, likely due to the compounds of interest not ionizing under the conditions used. However, the second-sourced sample of CDIQ165-N09 showed a clear peak with an m/z corresponding to an M+H adduct of the expected chemical whereas the ChemDiv sample showed no corresponding peak. Thus, the difference in observed responses is likely due to the expected chemical not being present in the ChemDiv sample at the expected concentration, possibly caused by degradation or handling of the library prior to our work. A few other compounds showed a slight deviation in fold-change of cleavage between the two preparations, likely due to differences in the final concentrations of the compounds. As the compound library preparation steps required liquid-handler pipetting of volumes in the low microliters, the limited precision of those transfers introduced deviations in the concentrations.

## Supporting information

**S1 Fig. Bistable amplicon sequences are capable of retaining "switching" capabilities by encoding.** The sequence above is representative of several sequences that were enriched early in the selection and contain a structure that appears to have two stable secondary structures. **(a)** secondary structure in which all the nucleotides are involved in forming the ribozyme; **(b)** an alternative secondary structure which leaves the 5' end free to anneal to the reverse transcription primer without disrupting the ribozyme structure.
(PDF)

**S2 Fig. Verification of compounds by mass spectrometry.** The elution time and m/z of the largest ion count peak matching expected adducts are shown for each of the compounds that occur in at least 4 out of 5 expected mixtures. Blue points indicate unambiguous assignments, red points are for compounds that overlap in elution time and m/z with at least one other compound, and magenta points show compounds that were not assigned an elution time. Data plotted here is contained in S1 Table.
(PDF)

**S3 Fig. Compound verification.** Each subplot shows the fold-change of cleavage of the sequences in the same library in response to two different formulations of purportedly the same compound. Error bars indicate the 95% confidence interval for each measurement based on the number of sequence reads; they are shown for sequences for which the lower-bound of the confidence interval is greater than 1.0 with either formulation.
(PDF)

**S1 Table. Table of all compounds.** Compounds used in this work. Each row includes: compound ID; SMILES; molecular weight; assignment to selection set A or B; V256 vectors containing the compound; maximum fold change observed when compound added in isolation at 10μM; minimum fold change observed when any vector containing the compound was added at 2μM; second source for compound, if any; mass spec identification (adduct, m/z, elution time, average ion count); number of false positives in mass spec identification at specified m/z and elution time; flag indicating, for each V256 group measured on mass spec, whether the compound was isolated.
(XLSX)

**S2 Table. Table of oligos used in manuscript.** Oligonucleotide and primer sequences used in this work, including ID, name, description, and sequence.
(XLSX)

**S3 Table. Summary of DRIVER selection rounds.** Table detailing conditions for each round of DRIVER selection including: template prefix, template concentration, whether it was done

manually or on the Tecan Freedom Evo (Robot), volume of transcription reaction, what compounds and at what total concentration were include, the splint-oligo used, the reverse-transcription volume, ligation volume, PCR primers used and the PCR volume and whether and how the round was cleaned up.
(XLSX)

**S4 Table. Table of oligo pools.** Oligonucleotide pools and members. The pool consisted of seven subgroups, named as shown in column 1. The pools with names starting with S7 were selected based on having a fold change of cleavage of at least 2.0 at round 95 of the selection. For each member of the pool, the sequences and ID are shown along with the pool name.
(XLSX)

**S5 Table. Table of biosensors.** Sequences of principal sensors isolated. Each distinct sequence that was measured against the set of 267 single compounds at 10μM and exhibited a fold change of cleavage of at least 2.0 is shown. These were then clustered into 150 groups (column 2) using the pattern of compounds to which the sequence responded. Columns 3–6 show the number of compounds that result in a fold change of cleavage of at least 2.0, 3.0, 5.0, or 8.0 respectively. The identity and fold change of the compounds which resulted in at least 2-fold change of cleavage are shown in column 7, and the sequence is shown in column 8 with spacing delineating the loops and stems of the expected secondary structure.
(XLSX)

**S6 Table. Table classifier model output.** Random forest classification of hits and misses. For each biosensor modeled, performance of the classifier is shown, including: number of compounds measured, number of compounds with at least 2-fold change of cleavage, the number of true negative classifications, the number of false-positives, the number of false-negatives, the number of true positives, the precision of the classifier, the true positive and negative rates of the classifier, and the area under the curve (AUC) of the receiver operating curve (ROC).
(XLSX)

**S1 File. All hits summary.** Comprehensive listing of each sensor identified, chart of fold change in the presence of each compound that affects it, and structures of those compounds in same format as Fig 10.
(PDF)

## Acknowledgments

The authors thank Dr. David Solow-Cordero of the Stanford High-Throughput Biosciences Center, Dr. Yuqin Liu of the Stanford ChemH Metabolomics Knowledge Center, and the Pehr Harbury lab for their help.

## Author Contributions

**Conceptualization:** Brent Townshend, Matias Kaplan, Christina D. Smolke.

**Data curation:** Brent Townshend.

**Formal analysis:** Brent Townshend, Matias Kaplan.

**Funding acquisition:** Christina D. Smolke.

**Investigation:** Brent Townshend, Matias Kaplan.

**Methodology:** Brent Townshend, Matias Kaplan, Christina D. Smolke.

**Project administration:** Christina D. Smolke.

**Resources:** Brent Townshend.

**Software:** Brent Townshend.

**Supervision:** Christina D. Smolke.

**Writing – original draft:** Brent Townshend, Matias Kaplan, Christina D. Smolke.

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
