## [Decision Letter · Decision Letter 0]

11 May 2022

PONE-D-22-07254Highly multiplexed selection of RNA aptamers against a small molecule libraryPLOS ONE

Dear Dr. Smolke,

Thank you for submitting your manuscript to PLOS ONE. After careful consideration, we feel that it has merit but does not fully meet PLOS ONE’s publication criteria as it currently stands. Therefore, we invite you to submit a revised version of the manuscript that addresses the points raised during the review process. As you will see below, reviewers were positive and raised issues that are mostly only minor. Several of their comments are directed toward ensuring that your manuscript is accessible to a broad audience and that it provides sufficient background / context for the reader unfamiliar with the field and with your previous Nature Comm. paper to see the merit in your approach. Please review their remarks carefully, ensure that there is sufficient detail in the Introduction to allow readers unfamiliar with DRIVER to understand the scope of the work and what aspects are new in this report, correct the errors noted, and address the remaining comments as you feel justified. I look forward to receiving your revised manuscript.

We look forward to receiving your revised manuscript.

Kind regards,

Floyd Romesberg

Academic Editor

PLOS ONE

Journal Requirements:

“This work was supported by the National Institutes of Health (grant to C.D.S.), National Science Foundation (graduate fellowships to M.K.), and Howard Hughes Medical Institute (Gilliam graduate fellowship to M.K.). C.D.S. is a Chan Zuckerberg Biohub investigator.”

4. Please ensure that you refer to Figure 6 in your text as, if accepted, production will need this reference to link the reader to the figure.

Reviewers' comments:

Reviewer's Responses to Questions

**Comments to the Author**

1. Is the manuscript technically sound, and do the data support the conclusions?

Reviewer #1: Yes

Reviewer #2: Yes

2. Has the statistical analysis been performed appropriately and rigorously? 

Reviewer #1: Yes

Reviewer #2: Yes

3. Have the authors made all data underlying the findings in their manuscript fully available?

Reviewer #1: Yes

Reviewer #2: Yes

4. Is the manuscript presented in an intelligible fashion and written in standard English?

Reviewer #1: Yes

Reviewer #2: Yes

5. Review Comments to the Author

Reviewer #1: Townshend et al. report a method for the selection of RNA-based biosensors to small molecule targets using a the previously reported in vitro evolution technique called DRIVER (de novo rapid in vitro evolution of RNA biosensors) which in the present work is applied to sets of mixtures of small molecules. The method allows for simultaneous selection of biosensors against multiple small molecule targets thereby increasing the efficiency of identification of active biosensors. The biosensor design consists of a hammerhead ribozyme in combination with an internal aptamer capable of recognizing small molecule targets. Binding of a small molecule target with affinity sufficient to cause a conformational change in the biosensor (involving components of the originally randomized loops) can be detected through the accompanying loss of self-cleavage enzymatic activity of the attached ribozyme. The selection is divided into several stages, which includes an initial selection for sequences capable of self-cleavage in the presence of two internal initially fully randomized loops, followed by two increasingly stringent cycles (each comprising multiple rounds) of positive and negative selections of sequences capable of exhibiting differential ribozyme activity in the presence of small molecule compounds.

The paper represents a proof of principle that highly multiplexed selections against a collection of small molecules are feasible, and as such, may be of interest to a wide audience of readers. At the same time, the paper could be improved by considering the following points.

Minor points:

• Figure 6 does (predicted structures of active and inactive switches to one of the representative small molecule compounds) not appear in the text of the manuscript.

Major points:

• For many of the hits with fold-change in cleavage of ≥2 summarized in the SI section (“SI Data – Hits.pdf”), there is no apparent common structural element among the small molecule compounds that attenuate the activity of the ribozyme. This makes it difficult to understand a plausible structural basis of recognition of such small molecules by the aptamer component of the biosensor. Although this is noted in a general manner in the discussion section, the authors should at least attempt to explain this observation in terms of the ability of the biosensors to recognize multiple structural motifs in one and the same RNA sequence.

• The number of rounds of selection required to achieve an enrichment for sequences with relatively modest difference in ribozyme cleavage efficiency is very large (95 rounds in total). This suggests relatively inefficient enrichment during each round. What is the reason for this inefficiency? Also, at what point does the work required to do multiple rounds of selection become more cumbersome than selections with fewer compounds at a time that may require fewer rounds of selection and less deconvolution? The authors may consider addressing these questions in the paper.

• It would be useful to state in the main text of the paper the actual concentrations of both the RNA library and the small molecules (both individually and collectively) in the mixtures that are incubated together during selections. This would allow the readers to consider the stoichiometry of the biosensor and small molecule targets that may drive the selections.

Reviewer #2: The manuscript by Townshend et al. (“Highly multiplexed selection of RNA aptamers against a small molecule library”) describes a mostly automated approach to significantly scale up a recently demonstrated method for parallelized selection of RNA aptamers against large pools of small molecule targets (DRIVER). The authors apply this DRIVER method to a pool of thousands of drug-like molecules, and such is the selection’s success that they spend a significant part of this manuscript deconvolving and identifying the hits, or sensor-target groupings. Such a large collection of hits—numbering in the hundreds—is a significant expansion of known interactions between RNA aptamers and small molecule targets, representing a rich dataset for future modeling work predicting interactions, sensitivities, targetability of arbitrary molecules, etc. This first scale-up of the DRIVER selection method also represents a useful starting point for refinements aimed at generating more sensitive and/or specific aptamers/sensors within this parallelized framework. The work is interesting, could easily generate follow-up studies, and would be suitable for publication one the following minor points are addressed.

1. Some cartoons depicting the specifics of how DRIVER and CleaveSeq work would be welcome, perhaps as part of Fig. 1, since these are so important for understanding this work. In particular, more introductory material about specifics like cleavage, ligation, sequence changes, etc. would be helpful for readers not already intimately familiar with these biosensors and methods.

2. Also worth briefly explaining in the intro is the means of translating the cleavage of the sensors presented here into in vivo control of gene expression, even if it’s just a one-sentence summary of ref. 18. This will help readers imagine applications in their own work.

3. Library section: the randomized regions of 30 + 8 nt seem pretty short—what would happen if this region were longer (or shorter)? Would that bias the eventual aptamers toward different sizes of target molecules, or would it simply not work as well because it’s constrained by the functionality of the ribozyme? Could be worth discussing a bit.

4. Library section: the mention of ligand interference with the tertiary interactions was very important for understanding how the sensors and selection works. Consider updating Fig. 1a to show how important that interaction is, and how it affects cleavage.

5. Library section: what purpose do the prefix and suffix serve? Are they worth mentioning here at all, as opposed to just in the methods or supplement?

6. Selection section: a small graphic showing how the primer binding sites achieve selective amplification could be helpful, perhaps combined with the structures in Fig. 1a.

7. Selection section: why do the rounds alternate between positive and negative selection? Or more specifically, what would happen if all rounds were positive selection? Some more context here would be helpful.

8. Selection section: why does the library concentration taper down to 1 µM only in the counter-selecting rounds? If it’s aimed at increasing sensitivity, why not decrease it quite a bit further in the last few positive selection rounds? And if not that, why taper it at all during the counter-selecting rounds?

9. Selection section: with so many rounds and with sequences harvested at the end, what is the risk that good aptamer sequences may have come and gone in the middle rounds? That is, enriched but then de-enriched due to binding-independent biases relating to amplification, purification, ligation, etc.?

10. CleaveSeq section, line 235: 1730 - 334 = 1396 is a lot of sequences that were inexplicably abundant! Any ideas for what's causing them to enrich?

11. Deconvolution section, line 271: the 217 number seems like a pretty solid lower bound if promiscuous sensors are excluded; perhaps it’s worth calling it “at least 217” in the abstract. Here the wording is fine because the next sentence makes that clear.

12. Deconvolution section: it might be worth mentioning just how much higher the ligand concentration is in Fig. 4 than in the mixed pools (2 µM total).

13. Sensitivity section, line 308: we thought (from the methods section) that it was 2 uM total for the 2560-compound mixtures. Please clarify this in all places where this concentration is mentioned, as it does indeed have an influence on sensitivity.

14. Enrichment profiles section: how many sequences that enriched and de-enriched are the authors not seeing? And are those strong binders, as indicated by early enrichment? What indications are there about the de-enrichment pressure being due to competition from better sequences (which would be fine) rather than due to binding/cleavage-independent logistical pressures (which would be bad), like amplification bias or the phantom primer-binding site described below? (De-enrichment during negative selection is fine and good, of course.)

15. Selectivity section, line 357: a bit more in depth and systematic analysis here would be good, especially a summary of trends observed in common high-binding groups, for example. The SI data 1 was not accessible.

16. Conclusion, line 406: Seems like cross-reactivity to probably-pretty-similar metabolites would be a recurring problem in this pathway-tracking strategy (but it certainly sounds possible and interesting in some cases!). This approach would seemingly be a good way to try to achieve that, but counterselection to the other metabolites would be a challenge; one would assume isolation of each metabolite's biosensor would be required before counterselection, right? Some more discussion about integrating these methods would be nice.

17. Conclusion, line 411: A note about the parallelizability of making these biosensors work in vivo as in refs. 17, 24, and 25 would be welcome here. (Thinking still about sensing multiple pathway members in real time.)

Conclusion, line 435: what criteria would decide which molecules to include in negative selection sets, and what effects would those criteria have on selectivity?

18. Figure 5: Interesting how the responses are similar across sensors; that is, sensor response is largely due to ligand identity, except for the bit in the main text about 125F11 (and maybe some others) that is a part of many different sequences' sets of target compounds. Therefore, it seems important that the authors or others can look for common chemical features linked to response characteristics, e.g. sensitivity, as noted above and in the Selectivity section. Again, be sure to fix the access issue with the SI data 1 file. Including a small depiction of the structures on these graphs would be an improvement.

19. Figure 6: maybe draw a boundary separating the switches from the nonswitches.

20. Figure 9: this is very hard to follow, making it hard to get any information out of it or determine the main message. E.g. are C and italic C different variables? Consider something simpler that clearly conveys (what we assume to be) the main message: that higher fold-change leads to lower cross-reactivity. Perhaps a series of histograms (one for each class of sensors capable of different maximal fold-changes with some compound) with "number of compounds eliciting a significant response" on the x-axis, or something similar.

6. PLOS authors have the option to publish the peer review history of their article (what does this mean?). If published, this will include your full peer review and any attached files.

Reviewer #1: No

Reviewer #2: No

---

## [Author Response · Author response to Decision Letter 0]

15 Jul 2022

Review Response to Editorial and Reviewer Comments:

We thank the reviewers for their valuable feedback and their constructive comments. Our point-by-point responses to the questions, comments and suggestions provided by the reviewers are provided in italics below. 

Journal Requirements

We have reformatted the manuscript to comply with the style requirements. 

We have checked both sections and confirm they refer to the same grant information (referencing NIH grant R01 GM 086663). We have included this information in the cover letter and removed it from the manuscript. 

“This work was supported by the National Institutes of Health (grant to C.D.S.), National Science Foundation (graduate fellowships to M.K.), and Howard Hughes Medical Institute (Gilliam graduate fellowship to M.K.). C.D.S. is a Chan Zuckerberg Biohub investigator.”

We note that you have provided additional information within the Acknowledgements Section that is not currently declared in your Funding Statement. Please note that funding information should not appear in the Acknowledgments section or other areas of your manuscript. We will only publish funding information present in the Funding Statement section of the online submission form. Please remove any funding-related text from the manuscript and let us know how you would like to update your Funding Statement. Currently, your Funding Statement reads as follows: “The funders had no role in study design, data collection and analysis, decision to publish, or preparation of the manuscript.” Please include your amended statements within your cover letter; we will change the online submission form on your behalf.

We have updated the language in this section and included amended statements in the cover letter to comply with your instructions. 

4. Please ensure that you refer to Figure 6 in your text as, if accepted, production will need this reference to link the reader to the figure.

We have revised our main text to cite Figure 6.

We have revised the manuscript text to comply.

We have reviewed the revised manuscript, and we do not cite any retracted papers. 

Reviewer #1 Comments: 

Townshend et al. report a method for the selection of RNA-based biosensors to small molecule targets using a the previously reported in vitro evolution technique called DRIVER (de novo rapid in vitro evolution of RNA biosensors) which in the present work is applied to sets of mixtures of small molecules. The method allows for simultaneous selection of biosensors against multiple small molecule targets thereby increasing the efficiency of identification of active biosensors. The biosensor design consists of a hammerhead ribozyme in combination with an internal aptamer capable of recognizing small molecule targets. Binding of a small molecule target with affinity sufficient to cause a conformational change in the biosensor (involving components of the originally randomized loops) can be detected through the accompanying loss of self-cleavage enzymatic activity of the attached ribozyme. The selection is divided into several stages, which includes an initial selection for sequences capable of self-cleavage in the presence of two internal initially fully randomized loops, followed by two increasingly stringent cycles (each comprising multiple rounds) of positive and negative selections of sequences capable of exhibiting differential ribozyme activity in the presence of small molecule compounds.

The paper represents a proof of principle that highly multiplexed selections against a collection of small molecules are feasible, and as such, may be of interest to a wide audience of readers. At the same time, the paper could be improved by considering the following points.

Minor points:

Figure 6 does (predicted structures of active and inactive switches to one of the representative small molecule compounds) not appear in the text of the manuscript.

We thank the reviewer for this comment. We have revised the main text of the manuscript to include a direct reference to Figure 6.

Major points:

For many of the hits with fold-change in cleavage of ≥2 summarized in the SI section (“SI Data – Hits.pdf”), there is no apparent common structural element among the small molecule compounds that attenuate the activity of the ribozyme. This makes it difficult to understand a plausible structural basis of recognition of such small molecules by the aptamer component of the biosensor. Although this is noted in a general manner in the discussion section, the authors should at least attempt to explain this observation in terms of the ability of the biosensors to recognize multiple structural motifs in one and the same RNA sequence.

We thank the reviewer for this feedback. We have included additional discussion in the main text regarding different structural patterns that we were able to discern from the existing data. We also include new discussion about specific sequences that can discriminate between different structures and information from our machine learning model that may serve as the basis for future experiments. We have further added a new figure (Figure 7) and included appropriate discussion.

The number of rounds of selection required to achieve an enrichment for sequences with relatively modest difference in ribozyme cleavage efficiency is very large (95 rounds in total). This suggests relatively inefficient enrichment during each round. What is the reason for this inefficiency?

We thank the reviewer for this comment. The reviewer rightly notes that DRIVER requires large numbers of rounds due to low enrichment efficiency. DRIVER was designed to select for small molecule sensors in solution by using changes in sequence (cleavage) of an RNA. During each round of selection, whether we are selecting for cleavers or non-cleavers, after sequences which cleave always or never cleave are depleted, we expect the background to cleave ~50% of the time regardless of target. This gives a maximum theoretical maximum of 2x enrichment per round. One finding of this current work is that even with this low enrichment rate, true biosensors still outcompete amplicons (Figure 8) and the large number of rounds can be efficiently executed with the automated DRIVER platform. We have expanded our discussion of the DRIVER process and specifically about enrichment mechanics and some of the enrichment mechanics we learned in carrying out DRIVER 5120. 

Also, at what point does the work required to do multiple rounds of selection become more cumbersome than selections with fewer compounds at a time that may require fewer rounds of selection and less deconvolution?

We thank the reviewer for this comment. We believe that lowering the number of molecules in a selection would not necessarily mean that fewer rounds of DRIVER could be run. Due to the low enrichment efficiencies, we still expect the majority of biosensors to appear in later rounds, since all biosensors start at a concentration of ~1 in 10^12 sequences. We have included in the main text a new discussion regarding a different set of drawbacks with the large selection numbers. These are specifically: 1) the need to deconvolve the large sets, and 2) difficulties in selecting highly selective biosensors without further negative selection. 

The authors may consider addressing these questions in the paper. It would be useful to state in the main text of the paper the actual concentrations of both the RNA library and the small molecules (both individually and collectively) in the mixtures that are incubated together during selections. This would allow the readers to consider the stoichiometry of the biosensor and small molecule targets that may drive the selections.

We thank the reviewer for this comment, and we have modified the main manuscript text to include more information regarding the ligand and RNA concentrations. 

Reviewer #2 Comments: 

The manuscript by Townshend et al. (“Highly multiplexed selection of RNA aptamers against a small molecule library”) describes a mostly automated approach to significantly scale up a recently demonstrated method for parallelized selection of RNA aptamers against large pools of small molecule targets (DRIVER). The authors apply this DRIVER method to a pool of thousands of drug-like molecules, and such is the selection’s success that they spend a significant part of this manuscript deconvolving and identifying the hits, or sensor-target groupings. Such a large collection of hits—numbering in the hundreds—is a significant expansion of known interactions between RNA aptamers and small molecule targets, representing a rich dataset for future modeling work predicting interactions, sensitivities, targetability of arbitrary molecules, etc. This first scale-up of the DRIVER selection method also represents a useful starting point for refinements aimed at generating more sensitive and/or specific aptamers/sensors within this parallelized framework. The work is interesting, could easily generate follow-up studies, and would be suitable for publication one the following minor points are addressed.

1. Some cartoons depicting the specifics of how DRIVER and CleaveSeq work would be welcome, perhaps as part of Fig. 1, since these are so important for understanding this work. In particular, more introductory material about specifics like cleavage, ligation, sequence changes, etc. would be helpful for readers not already intimately familiar with these biosensors and methods.

We thank the reviewer for this comment and have revised the manuscript text to include a more thorough explanation of DRIVER and a diagram in Figure 1 for readers to refer to. 

2. Also worth briefly explaining in the intro is the means of translating the cleavage of the sensors presented here into in vivo control of gene expression, even if it’s just a one-sentence summary of ref. 18. This will help readers imagine applications in their own work.

We thank the reviewer for this comment and have updated the text to include an explanation of in vivo use of our biosensors. 

3. Library section: the randomized regions of 30 + 8 nt seem pretty short—what would happen if this region were longer (or shorter)? Would that bias the eventual aptamers toward different sizes of target molecules, or would it simply not work as well because it’s constrained by the functionality of the ribozyme? Could be worth discussing a bit.

We thank the reviewer for this comment, based on pilot studies, our library included loops with size 30 on one stem and loops with either 4,5,6,7 or 8 random nucleotides on the opposite loop. We believe that a systematic study of the effect of different loop lengths on the selection should be pursued in later studies. We have noted this in our discussion. 

4. Library section: the mention of ligand interference with the tertiary interactions was very important for understanding how the sensors and selection works. Consider updating Fig. 1a to show how important that interaction is, and how it affects cleavage.

We thank the reviewer for this comment. We have revised Figure 1a to illustrate this mechanism more clearly. 

5. Library section: what purpose do the prefix and suffix serve? Are they worth mentioning here at all, as opposed to just in the methods or supplement?

We thank the reviewer for this comment. The prefix and suffix are used for the PCR of the selected products and the prefix is used to distinguish between cleaved and uncleaved molecules (following the ligation of a new prefix to cleaved parts). We have revised the manuscript text to include a description of the purpose of these elements in the DRIVER process. 

6. Selection section: a small graphic showing how the primer binding sites achieve selective amplification could be helpful, perhaps combined with the structures in Fig. 1a.

We thank the reviewer for this comment and have revised Figure 1 to incorporate this information. 

7. Selection section: why do the rounds alternate between positive and negative selection? Or more specifically, what would happen if all rounds were positive selection? Some more context here would be helpful.

We thank the reviewer for this comment. The purpose of the alternating rounds of positive and negative selection is to ensure that we are enriching sequences capable of switching. For our purposes this means our sequences must cleave in the absence of ligand (negative selection) and not cleave in the presence of ligand (positive selection). If all rounds were positive selection we would select for sequences which self-cleave efficiently independently of the presence of the target molecule. We have revised the manuscript text to clarify this point. 

8. Selection section: why does the library concentration taper down to 1 µM only in the counter-selecting rounds? If it’s aimed at increasing sensitivity, why not decrease it quite a bit further in the last few positive selection rounds? And if not that, why taper it at all during the counter-selecting rounds?

We thank the reviewer for this comment. The library concentration was reduced to 1 µM in both the positive and negative rounds. We have revised the manuscript text to clarify this point. The reduction in concentration during the latter part of the selection was aimed at increasing sensitivity; however, it is difficult to know a priori where the optimal tradeoff between loss of binders and specific enrichment of strong binders occurs. We chose to be conservative in this study, as the primary goal was to demonstrate the identification of sensors to as many distinct molecules as possible. Subsequent enrichment could be performed on selected hits with lower library concentrations to enrich for more sensitive binders if that is a desired outcome. We have added in additional text to the manuscript to clarify these choices.

9. Selection section: with so many rounds and with sequences harvested at the end, what is the risk that good aptamer sequences may have come and gone in the middle rounds? That is, enriched but then de-enriched due to binding-independent biases relating to amplification, purification, ligation, etc.?

We thank the reviewer for this comment. While the scenario described by the reviewer is possible, if a sequence was enriched in the first place, it is likely that the only reason that sequence would be subsequently de-enriched is if another sequence with a higher “fitness”, such as a better binder, becomes dominant and decreases the relative concentration of other sequences. That would only occur if the bulk of the library has increased in fitness – that is, the whole library is responding to the ligand(s). Other binding-independent biases, if present, would affect the initial enrichment as well – the primary proof that amplicons are not dominating is the result that biosensor sequences do become enriched. 

10. CleaveSeq section, line 235: 1730 - 334 = 1396 is a lot of sequences that were inexplicably abundant! Any ideas for what's causing them to enrich?

We thank the reviewer for this comment. The set of 1730 was composed of sequences chosen for various reasons. In particular, the 200 most abundant sequences that were not biosensors were synthesized to better understand these sequences – that choice was independent of the relative fraction of switching vs non-switching sequences. In addition, we included 210 sequences that were modified versions of sequences identified as biosensors during the selection to understand better the effect of introducing certain changes in the stem loops. We also included sequences identified as possible hits during preliminary runs of the selection. In terms of the abundant sequences that were not biosensors, we observed that many of these sequences have multiple conformations, some of which allow for cleavage and some that do not. These results indicate that under the selective pressures applied in this study a large enough fraction would survive in both the cleaved and the uncleaved selection steps to continue. These types of sequences are the reason that the enrichment/cycle is limited to approximately two since a sequence that adopts a switching or a non-switching conformation at approximately equal frequency will survive 50% of the time in both negative and positive selection rounds. Another version of the sequences contains a complementary sequence to the HHRz stem III within the randomized loop. This sequence allows the biosensor to escape our stop oligo by recapitulating the HHRz secondary structure. We have modified the manuscript to include further details about these sequences. Future studies can focus on how the remainder of these biosensors work to try and limit the number of non-ligand-responsive sequences we observe. 

11. Deconvolution section, line 271: the 217 number seems like a pretty solid lower bound if promiscuous sensors are excluded; perhaps it’s worth calling it “at least 217” in the abstract. Here the wording is fine because the next sentence makes that clear.

We thank the reviewer for this comment. We have modified the abstract as suggested by this reviewer.

12. Deconvolution section: it might be worth mentioning just how much higher the ligand concentration is in Fig. 4 than in the mixed pools (2 µM total).

We thank the reviewer for this comment. We have modified the caption of Figure 4 to include this detail. 

13. Sensitivity section, line 308: we thought (from the methods section) that it was 2 uM total for the 2560-compound mixtures. Please clarify this in all places where this concentration is mentioned, as it does indeed have an influence on sensitivity.

We thank the reviewer for this comment. The last several rounds were reduced to 1µM as discussed above. We have modified the manuscript text to clarify this point.

14. Enrichment profiles section: how many sequences that enriched and de-enriched are the authors not seeing? And are those strong binders, as indicated by early enrichment? What indications are there about the de-enrichment pressure being due to competition from better sequences (which would be fine) rather than due to binding/cleavage-independent logistical pressures (which would be bad), like amplification bias or the phantom primer-binding site described below? (De-enrichment during negative selection is fine and good, of course.)

We thank the reviewer for this comment. Since we can only capture through sequencing a relatively small random subsample of the sequences present, it is not feasible to track the progress of sequences throughout the process unless they exceed ~1 part-per-billion (and more practically, 1 ppm). As we start with approximately one molecule per sequence, most of the selection process is not directly observable at an individual sequence level. It would be interesting to model the true diversity of biosensors that we are not seeing, but that is outside of the scope of the experiments in this manuscript. 

One of the strongest pieces of evidence for the selection working to enrich biosensors is that in the first ~20 rounds, amplicon sequences make up an overwhelming majority (Figure 7, top row) of sequences in those rounds. However, as selection continues the proportion of amplicons to true biosensors greatly diminishes. Furthermore, once the non-switching background reaches approximately 50% cleavage, but the pool does not show bulk switching activity, the fitness of the pool as a whole is constant and enrichment/de-enrichment pressure on any sequence is dependent only on the fitness of that sequence. Thus, biosensors present at the end of the selection were enriched throughout the selection. 

15. Selectivity section, line 357: a bit more in depth and systematic analysis here would be good, especially a summary of trends observed in common high-binding groups, for example. The SI data 1 was not accessible.

We thank the reviewer for this comment. We have modified the manuscript text in this section to include additional discussion of the trends observed in common high-binding groups and similarities and differences between the target compounds. We have confirmed the SI data 1 is included with the submission. 

16. Conclusion, line 406: Seems like cross-reactivity to probably-pretty-similar metabolites would be a recurring problem in this pathway-tracking strategy (but it certainly sounds possible and interesting in some cases!). This approach would seemingly be a good way to try to achieve that, but counterselection to the other metabolites would be a challenge; one would assume isolation of each metabolite's biosensor would be required before counterselection, right? Some more discussion about integrating these methods would be nice.

We thank the reviewer for this comment. We agree with the reviewer that cross-reactivity to similar metabolites is a common challenge in biosensor approaches (in general, not necessarily specific to RNA-based biosensors) for metabolic pathway tracking. To achieve desired specificities, counterselections would be required against similar metabolites that the researcher wants to distinguish against. The biosensors would not need to be isolated necessarily; however, the products of a highly multiplexed selection could then be used as input to multiple targeted selections, each one desired to tune for desired properties or specificities. We have modified the discussion in this section to highlight some of these points. 

17. Conclusion, line 411: A note about the parallelizability of making these biosensors work in vivo as in refs. 17, 24, and 25 would be welcome here. (Thinking still about sensing multiple pathway members in real time.) Conclusion, line 435: what criteria would decide which molecules to include in negative selection sets, and what effects would those criteria have on selectivity?

We thank the reviewer for this comment. We have added an additional line about the parallelizability in this section of the manuscript text. We think that the technical details of what molecules to include in a negative selection need to be further studied through systematic experiments. Broadly, we believe that aptamers selected against a large mix of small molecules may be over-enriched for shared chemical substructures within the small molecule set. Hence, if we want to create more specific aptamers we need to select against those shared substructures. Additionally, we could envision wanting to include known cellular metabolites within the negative selection mix to mirror the in-vivo environment as closely as possible. 

18. Figure 5: Interesting how the responses are similar across sensors; that is, sensor response is largely due to ligand identity, except for the bit in the main text about 125F11 (and maybe some others) that is a part of many different sequences' sets of target compounds. Therefore, it seems important that the authors or others can look for common chemical features linked to response characteristics, e.g. sensitivity, as noted above and in the Selectivity section. Again, be sure to fix the access issue with the SI data 1 file. Including a small depiction of the structures on these graphs would be an improvement.

We thank the reviewer for this comment and have added in an additional section in the discussion highlighting some patterns we saw in the data about how chemical substructure influences sensing sensitivity and specificity. We have ensured that SI data file 1 is submitted and accessible. We have also modified the graphs to incorporate depictions of the structures as suggested by this reviewer. 

19. Figure 6: maybe draw a boundary separating the switches from the nonswitches.

We thank the reviewer for this comment and have adapted the figure to address the concerns. We have reformatted Figure 6 to more clearly show the ranges of activity for all of the sequences as well as provide more summary statistics.

20. Figure 9: this is very hard to follow, making it hard to get any information out of it or determine the main message. E.g. are C and italic C different variables? Consider something simpler that clearly conveys (what we assume to be) the main message: that higher fold-change leads to lower cross-reactivity. Perhaps a series of histograms (one for each class of sensors capable of different maximal fold-changes with some compound) with "number of compounds eliciting a significant response" on the x-axis, or something similar.

We thank the reviewer for this comment. The C and italic C are the same variables - there is no italic C so there may be an issue with rendering of text in the formatted figure. While we appreciate the reviewer’s input on the figure we have chosen to not alter this figure as we believe it presents the information we were seeking to convey in its current form.

---

## [Editor Report · Decision Letter 1]

8 Aug 2022

Highly multiplexed selection of RNA aptamers against a small molecule library

PONE-D-22-07254R1

Dear Dr. Smolke,

We’re pleased to inform you that your manuscript has been judged scientifically suitable for publication and will be formally accepted for publication once it meets all outstanding technical requirements.

Kind regards,

Floyd Romesberg

Academic Editor

PLOS ONE

Additional Editor Comments (optional):

The authors have thoughtfully addressed the reviewers' comments and have revised the text to address the minor points raised by the reviewers. Additional data and discussion have been added in an attempt to address the different structural motifs among the small molecule attenuators of the ribozyme's activity. Any further detail would be beyond the scope of the current work. With the many changes made and especially with the revisions that more clearly explain DRIVER and the relationship between the current manuscript and the former report of the method, this work is now suitable for publication in PLoS ONE.

---

## [Editor Report · Acceptance letter]

6 Sep 2022

PONE-D-22-07254R1 

Highly multiplexed selection of RNA aptamers against a small molecule library 

Dear Dr. Smolke:

I'm pleased to inform you that your manuscript has been deemed suitable for publication in PLOS ONE. Congratulations! Your manuscript is now with our production department. 

Kind regards, 

on behalf of

Dr Floyd Romesberg 

Academic Editor

PLOS ONE